# Specialized transendothelial dendritic cells mediate thymic T-cell selection against blood-borne macromolecules

Elisabeth H. Vollmann[1,6], Kristin Rattay [1,2], Olga Barreiro[1], Aude Thiriot [1], Rebecca A. Fuhlbrigge [1], Vladimir Vrbanac[3,7], Ki-Wook Kim [4,8], Steffen Jung [4], Andrew M. Tager[3] & Ulrich H. von Andrian [1,5 ✉]

T cells undergo rigorous selection in the thymus to ensure self-tolerance and prevent autoimmunity, with this process requiring innocuous self-antigens (Ags) to be presented to thymocytes. Self-Ags are either expressed by thymic stroma cells or transported to the thymus from the periphery by migratory dendritic cells (DCs); meanwhile, small blood-borne peptides can access the thymic parenchyma by diffusing across the vascular lining. Here we describe an additional pathway of thymic Ag acquisition that enables circulating antigenic macromolecules to access both murine and human thymi. This pathway depends on a subset of thymus-resident DCs, distinct from both parenchymal and circulating migratory DCs, that are positioned in immediate proximity to thymic microvessels where they extend cellular processes across the endothelial barrier into the blood stream. Transendothelial positioning of DCs depends on DC-expressed $CX_3CR1$ and its endothelial ligand, $CX_3CL1$, and disrupting this chemokine pathway prevents thymic acquisition of circulating proteins and compromises negative selection of Ag-reactive thymocytes. Thus, transendothelial DCs represent a mechanism by which the thymus can actively acquire blood-borne Ags to induce and maintain central tolerance.

[1] Department of Immunology & HMS Center for Immune Imaging, Harvard Medical School, Boston, MA 02115, USA. [2] Institute of Pharmacology, Biochemical Pharmacological Center, University of Marburg, Marburg, Germany. [3] Massachusetts General Hospital, Boston, MA, USA. [4] Department of Immunology, Weizmann Institute of Science, Rehovot, Israel. [5] Ragon Institute of MGH, MIT and Harvard, Cambridge, MA, USA. [6] Present address: Merck Research Laboratories, Boston, MA 02115, USA. [7] Present address: Massachusetts General Hospital, Humanized Immune System Mouse Program (HISMP), Boston, MA 02114, USA. [8] Present address: Department of Pharmacology and Regenerative Medicine, University of Illinois College of Medicine, Chicago, IL 60612, USA. ✉email: uva@hms.harvard.edu

The thymus periodically recruits bone marrow (BM)-derived lymphoid progenitors through microvessels at the junction between the inner medulla and outer cortex. These progenitors then develop into T cells by undergoing a sequence of differentiation steps during which they rearrange T cell receptor (TCR) gene segments to generate a broad repertoire of antigen (Ag) specificities[1,2]. A series of concomitant quality control steps deletes non-self-restricted and autoreactive thymocytes or redirects them to a regulatory lineage[2]. Positive selection, which enriches for self-restricted thymocytes, is thought to take place in the cortex, while negative selection, the process that removes autoreactive T cells, is believed to occur preferentially in the medulla and cortico-medullary junction (CMJ)[3–5]. This selection process relies on the presentation of diverse innocuous self-Ags from every tissue in the body and requires orchestrated interactions between motile thymocytes and Ag-presenting dendritic cells (DCs) and/or thymic epithelial cells (TECs)[2,5,6].

Imaging studies have characterized the complex dynamics of intrathymic cell–cell interactions using multi-photon microscopy[7–10]. These experiments have usually been performed ex vivo rather than in living animals because intravital imaging of the intact thymus in its native location is technically difficult. The thoracic thymus contacts the lung and heart, which are constantly moving, creating motion artifacts that make time- and space-resolved imaging of intrathymic processes challenging. Similarly, ectopic thymi that are found in the neck region of mice are poorly suitable for intravital microscopy (IVM) approaches, such as multi-photon intravital microscopy (MP-IVM), as their occurrence, location, and number are highly variable[11]. Thus, most investigators have resorted to imaging excised thymi that were sectioned and maintained in vitro[8,9]. These ex vivo imaging approaches have mainly focused on the T cell component of the thymus, whereas little is known about the dynamic behavior of other thymic constituents, including DCs. Moreover, studies in excised tissues cannot visualize the thymic interface with the blood circulation, which serves as an essential conduit through which lymphoid progenitors, migratory DCs, and free Ags gain access to the thymus. The processes that make diverse Ags bioavailable in the thymus are important to understand because they are essential to establish self/non-self-discrimination by nascent T cells to induce central tolerance[2,5,6].

Many central tolerance-inducing self-Ags, such as ubiquitously expressed proteins, are produced by the cellular constituents of the thymus itself. Certain tissue-restricted Ags are also expressed ectopically by medullary TECs (mTECs)[12], which can directly present these Ags to thymocytes[13] as well as make them available for presentation by thymic DCs[14]. There are also Ags to which tolerance is needed, but that are not expressed by thymus-resident cells. Such Ags, e.g., material from commensal flora, must be delivered to the thymus from the periphery[15]. One delivery mechanism involves migratory DCs that acquire Ags in peripheral tissues and then access the blood and carry their cargo to the thymus[16,17], another mechanism applies to blood-borne small antigenic peptides, which can enter the thymus by diffusing directly from the blood[18].

To date, it has been unclear whether there is an active thymic delivery mechanism for blood-borne macromolecules that are too large to diffuse passively from the intra- to the extravascular thymic compartment. To address this question, here we study the thymus-blood interface by transplanting thymi under the kidney capsule of mice and subjecting the recipients to MP-IVM. We identify a discrete subset of thymic DCs that colocalize with microvascular endothelial cells (ECs) and directly monitor the flowing blood in murine and human thymi. We name these cells transendothelial DCs (TE-DCs) because they extend protrusions across the endothelium into the vascular lumen. Most TE-DCs are $CD11c^+CD8\alpha^-CD11b^+$ and require the $CX_3CR1–CX_3CL1$ pathway to maintain their bi-compartmental positioning. We show that TE-DCs capture macromolecules from the bloodstream and shuttle this antigenic material into the thymus for tolerance induction in developing T cells. This route of thymic antigen delivery promotes the formation of central tolerance against blood-borne macromolecules that are not expressed within the thymus itself and thereby complements the thymic antigen repertoire for the induction of central tolerance and the prevention of autoimmunity.

## Results

**Characterization of transplanted thymi for in vivo imaging.** To circumvent the challenges associated with imaging the thoracic thymus by MP-IVM, we grafted embryonic (E18) thymi under the left kidney capsule of 4-week-old male recipient mice. One month after transplantation, transplanted thymi (TT) were well vascularized and organized into cortical and medullary zones, as determined by immunohistochemistry and flow cytometry (Fig. 1a, b, and Supplementary Fig. 1a, b). Notably, the average distance between the capsule and medulla was reduced by more than half in TT versus endogenous thymi (ET; Fig. 1c), presumably owing to the overall smaller size and reduced cellularity of the transplanted organs (Supplementary Fig. 1c). However, the ratio between cortical thymic epithelial cells (cTECs) and mTECs, as well as the relative frequency of thymocyte subsets, was identical in TT and ET, indicating similar composition and function (Fig. 1b and Supplementary Fig. 1d). Also, in both ET and TT the distribution of DCs (defined as $CD11c^+Flt3^+c-Kit^+CD26^+$) was strongly biased toward the medulla and CMJ, whereas DCs were sparse in the cortex (Fig. 1d and Supplementary Fig. 1b). DC subset frequencies were also similar in TT and ET, with plasmacytoid DCs (pDCs) and $CD8\alpha^+CD11b^-$ DCs being over-represented relative to the $CD8\alpha^-CD11b^+$ DC subset, which constituted the majority of DCs in secondary lymphoid organs, such as the spleen (Fig. 1e).

Having determined that TT closely resembles their ET counterparts in architecture and cellular composition, we asked whether circulating immature DCs home to TT, as has been shown for ET[16]. Indeed, TT readily recruited adoptively transferred immature (but not LPS-matured) DCs (Fig. 1f), requiring the same traffic molecules as ET (Fig. 1g-j). Specifically, DC homing was abrogated by DC treatment with pertussis toxin (PTX), which blocks chemoattractant signaling through $G\alpha_i$-coupled receptors, such as CCR9[17], triggering intravascular arrest (Fig. 1g). The firm arrest is thought to depend on integrins, particularly DC-expressed VLA-4 (a.k.a., CD49d–CD29 or $\alpha 4\beta 1$ integrin), which binds constitutively expressed VCAM-1 in thymic microvessels. Blocking either receptor virtually abolished DC homing to both ET and TT, but not the spleen (Fig. 1h, i), consistent with the previous studies[16]. Homing was also blocked by a monoclonal antibody (mAb) against endothelial P-selectin (Fig. 1j), which is known to mediate rolling in thymic microvessels by interacting with glycosylated DC-expressed PSGL-1[16].

We conclude that TT faithfully mirrored ET with the exception of a more superficial location of medullary regions in TT. The latter feature presents a distinct advantage for MP-IVM because thymic tissue, in particular the cortex, is quite opaque to visible light, which makes it difficult to image the medulla and CMJ where most thymic DCs reside[9]. In our hands, the maximal penetration depth that can be routinely achieved by MP-IVM in intact murine thymi is ~200 µm. Only ~21% of thoracic ET had medullary regions located within this detection range, whereas

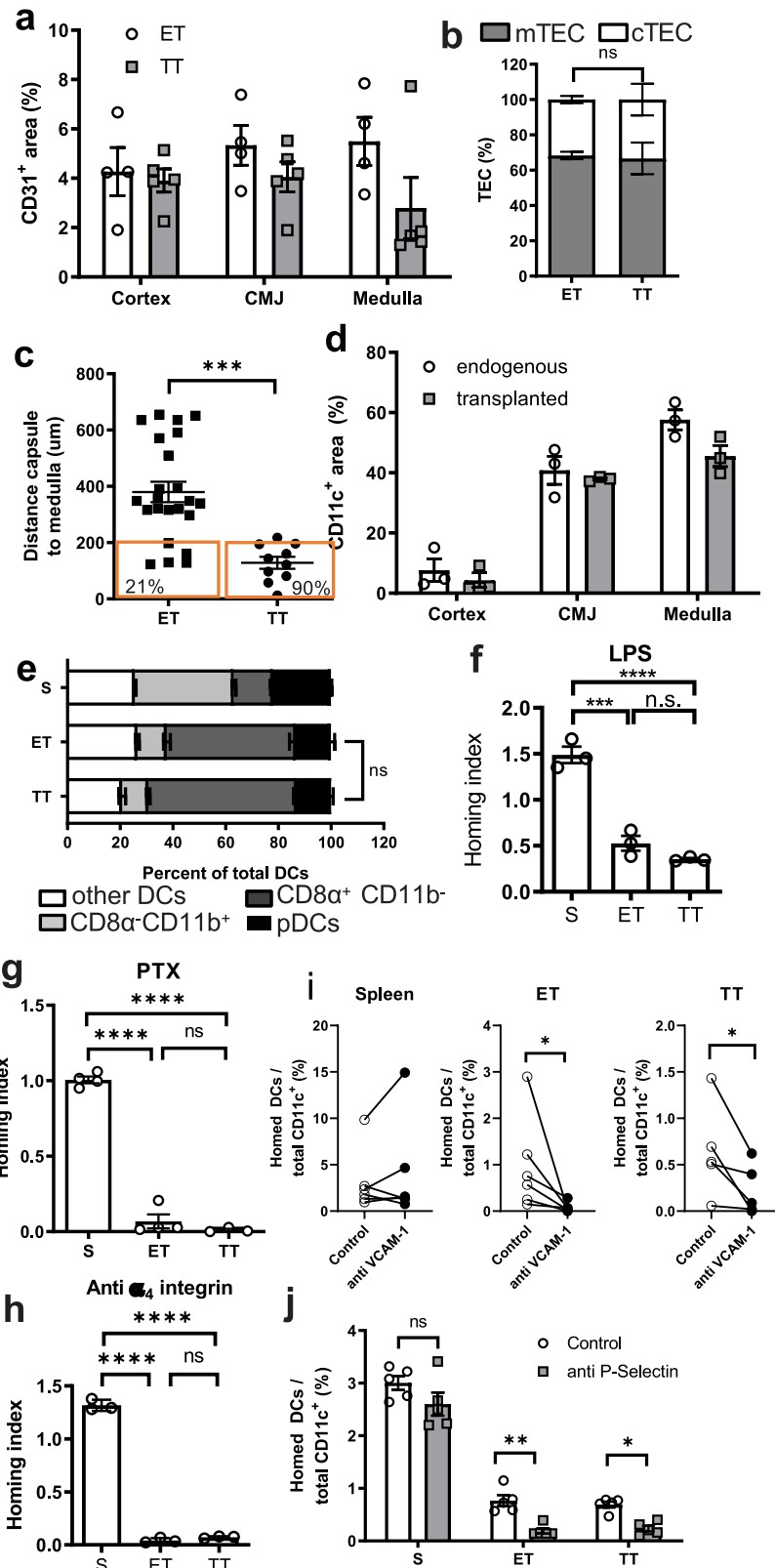

>90% of medullas in TT were accessible for MP-IVM visualization (Fig. 1c).

**Motile behavior of homed and resident DCs in the thymus.** Having determined that thymus transplantation under the kidney capsule makes MP-IVM feasible without disrupting normal thymic architecture or function, we sought to evaluate the in vivo behavior of thymic DCs. Based on their site of origin, thymic DCs can be subdivided into an endogenous population that originates from progenitors within the thymus and a population of homed cells that enter the thymus from the blood as fully differentiated DCs. The latter subset includes DCs that collect and transport Ags from peripheral tissues[16,17,19]. To discriminate between these two populations, four weeks old mice were lethally irradiated and

**Fig. 1 Characterization of endogenous and transplanted thymi.** Fetal thymi were transplanted under kidney capsules of recipient mice and endogenous thymi (ET) and transplanted thymi (TT) were analyzed 4–9 weeks later. **a** Endothelial cell density in the cortex, the cortico-medullary junction (CMJ), and medulla were assessed in frozen sections of TT (closed bars) and ET (open bars) after staining for CD31 and region-specific markers (Supplementary Fig. 1a); $n = 5$ experiments. **b** Relative frequencies of medullary (mTECs; filled bar) and cortical thymic epithelial cells (cTECs; empty bar white) in TT and ET were assessed by flow cytometry. $n = 5$. **c** The shortest distance between capsule and medullary border in TT ($n = 10$) and ET ($n = 23$; Student's two-tailed, unpaired t test, $P = 0.0001$). **d** Dendritic cell (DC) density in cortex, CMJ and medulla in frozen sections of TT (closed bars) and ET (open bars) after staining for CD11c and region-specific markers (Supplementary Fig. 1b); $n = 5$ experiments. **e** DC subset composition in spleen (S), ET, and TT. Data were pooled from two independent experiments with $n = 10$ mice/each. **f–j** Recruitment of circulating DCs to spleen (S), TT and ET was compared in competitive homing assays. A 1:1 mixture of differentially labeled treated and control DCs was injected IV into mice that were previously transplanted with a fetal thymus. The homing index (i.e., the ratio of treated:control DC corrected for input ratio) was determined after 18 h by FACS in single-cell suspensions of each recipient tissue. Homing indices were markedly reduced in ET and TT, but not in spleen after **f** LPS-induced DC maturation, **g** inhibition of chemokine receptor signaling with pertussis toxin (PTX), or **h** DC treatment with a blocking anti-$\alpha_4$ integrin mAb. Results were pooled from two independent experiments, $n = 10$ mice/group. **i** and **j** The role of endothelial adhesion molecules in homing of adoptively transferred DCs was assessed after 18 h by comparing DC accumulation in control mice and animals treated with blocking mAb to VCAM-1 (**i**) or P-selectin (**j**). Pooled data from two experiments, $n = 5$ mice/group. **f, g, i** ****$P < 0.0001$, ***: $P = 0.0002$. **h** **$P = 0.0069$, *$P = 0.0364$. **j** *$P = 0.0369$ (ET) and $P = 0.022$ (TT).

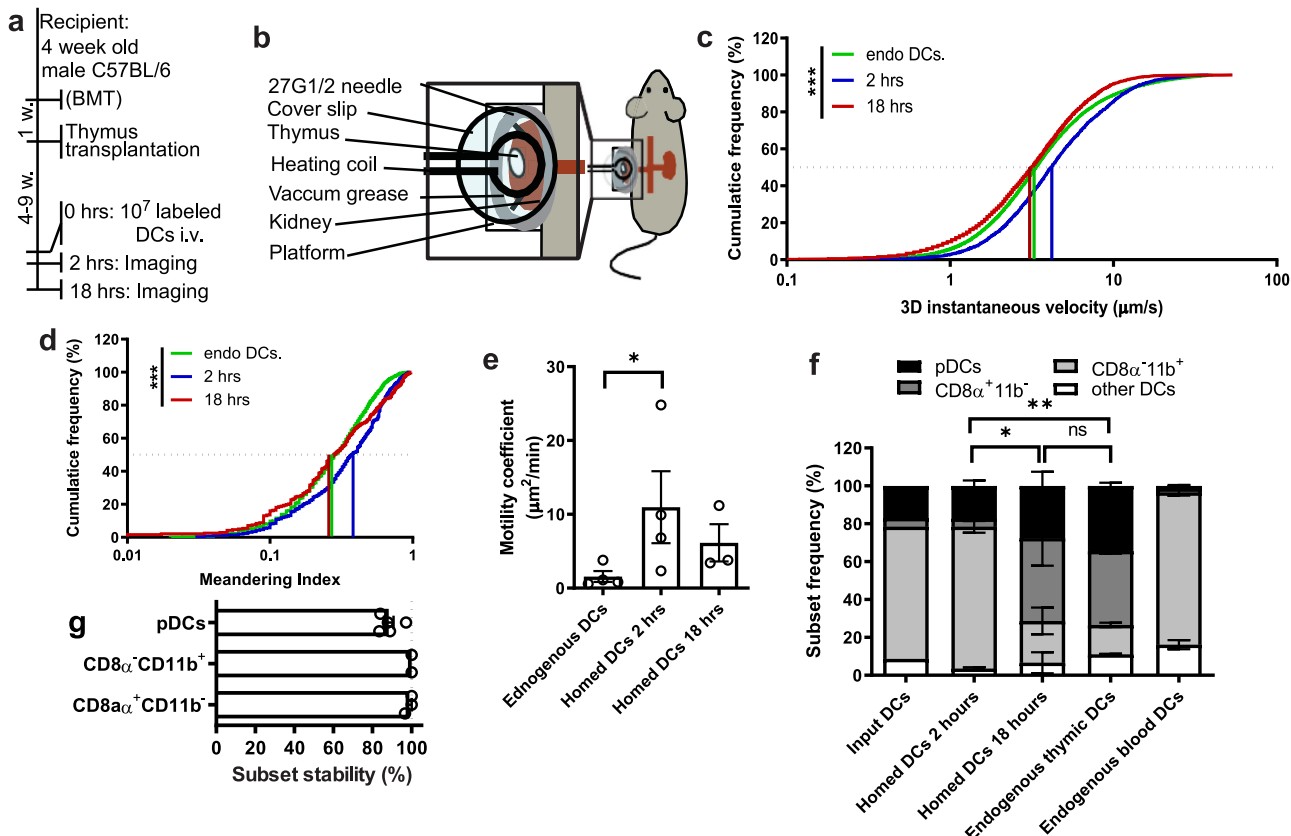

**Fig. 2 Analysis of dendritic cells in endogenous and transplanted thymi.** **a** Experimental protocol for thymic MP-IVM experiments. BMT: bone marrow transplantation. **b** Schematic drawing of the experimental setup to image transplanted thymi by MP-IVM. **c** 3D instantaneous velocity and **d** meandering index of thymus-resident DCs and homed DCs at 2 hours and 18 hours post transfer. $n = 3$ independent experiments. **e** Motility coefficients of endogenous and homed DCs. $n = 4$ independent experiments for 2 hrs and $n = 3$ independent experiments for 18 hrs time points. **f** Subset composition of resident thymic DCs and homed DCs 2 and 18 h post transfer. $n = 2$ experiments with 10 mice/group. **g** FACS-sorted splenic DC subsets were fluorescently labeled and injected IV into recipient mice. 18 h later, subset-specific markers on homed DCs in recipient thymi were analyzed by flow cytometry. DC subsets were sorted as follows: pDCs: CD11c$^+$ PDCA1$^+$ B220$^+$; CD8$\alpha^+$CD11b$^-$ DCs: CD11c$^{high}$ PDCA1$^-$ B220$^-$ CD8$\alpha^+$ CD11b$^-$; CD8$\alpha^-$CD11b$^+$ DCs: CD11c$^{high}$ PDCA1$^-$ B220$^-$ CD8$\alpha^+$ CD11b$^-$. $n = 5$ mice/group. Bars and error bars represent mean ± SEM. Statistical comparisons were performed using two-way ANOVA unless stated otherwise. *$P = 0.0428$ (**e**) and $P = 0.0142$ (**f**); **$P = 0.0026$, ***$P < 0.001$; n.s. not significant. DCs dendritic cells, pDCs plasmacytoid dendritic cells. See also Supplementary Fig. 1 and Supplementary Movies 1 and 2.

transplanted with mixed BM from CD11c-YFP[20] and WT donors at a 1:1 ratio (Fig. 2a). The transplanted animals generated a dense network of thymic DCs among which a fraction could be visualized owing to their expression of yellow fluorescent protein (YFP). The rationale for diluting CD11c-YFP donor BM with WT BM was to adjust the frequency of fluorescent DCs to a density at which individual cells could be readily identified and tracked; this was not feasible when all thymic DCs carried YFP because fluorescent signals from individual cells became confluent and did not permit a clear delineation of cell–cell borders. The BM chimeric animals were rested for one week and then received thymus transplantation followed by MP-IVM 4–9 weeks later. To

introduce homed DCs, some transplanted animals were injected IV with purified splenic DCs that were fluorescently labeled with 7-amino-4-chloromethylcoumarin (CellTracker™ Blue, CMAC), a fluorophore that is spectrally distinct from YFP expressed in resident DCs.

To perform MP-IVM, the animals were anesthetized and thymus-bearing kidneys were gently exteriorized without interrupting blood flow and immobilized on a Styrofoam platform with 27G1/2 needles distally to the graft (Fig. 2b). The exposed organ was submerged in normal saline to prevent drying and covered with a glass coverslip attached to a stereotactic holder. A thermocouple was placed next to the thymus to monitor local temperature, which was maintained at 37 °C with a heating coil. Endogenous (YFP+) and homed (CMAC+) thymic DCs were imaged at either 2 h or 18 h after DC injection.

In all, 2 h after transfer, homed DCs were more motile than endogenous DCs, as evidenced by significantly greater 3D instantaneous velocity (Fig. 2c; $P < 0.001$), meandering index (Fig. 2d; $P < 0.001$), and motility coefficient (Fig. 2e; $P < 0.05$). However, at 18 h post transfer the homed DCs displayed a motile behavior reminiscent of that of endogenous DCs (Fig. 2c–e and Supplementary Movies 1 and 2). This change could reflect a cell-intrinsic reduction in motility after prolonged intrathymic dwell time or a change in the homed DC subset composition, whereby highly motile cells that had accessed the thymus more quickly may have been replaced by slower moving cells that predominated at later time points. To address this issue, we compared the composition of homed DC subsets at 2 h and 18 h post transfer to that of ET DCs as well as endogenous blood-borne DCs and the initial input population of adoptively transferred DCs. After 2 h, homed DCs reflected the (spleen derived) input population and contained a predominant fraction of CD11c+CD8α−CD11b+ DCs, whereas CD11c+CD8α+CD11b− cells and pDCs were underrepresented as compared to ET DCs (Fig. 2f). By contrast, at 18 h post transfer, the frequency of the CD11c+CD8α+CD11b− subset (and to some extent pDCs) had increased at the expense of the CD11c+CD8α−CD11b+ subset, resulting in a composition of homed DCs similar to that of ET DCs. This gradual change in composition was not due to phenotypic plasticity, as homed DCs remained phenotypically stable when individual purified subsets were adoptively transferred (Fig. 2g). Thus, the shift in homed DC subset composition was either a consequence of preferential retention or survival (or both) of pDCs and CD8α+CD11b− DCs as compared with CD8α−CD11b+ DCs in the thymic microenvironment.

**Identification of transendothelial DCs.** Although performing MP-IVM in TT in which the microvasculature was visualized after IV injection of high molecular weight TRITC-dextran, we noted that ~1/4 of thymic DCs were conspicuously positioned in the immediate vicinity (<1 μm distance) to microvessels (Fig. 3a, b and Supplementary Movie 3). This observation was confirmed by a parallel immunohistological analysis of frozen thymic sections, which indicated that some perivascular DCs extended protrusions across the endothelial barrier into the vessel lumen (Fig. 3c). To pursue this observation more rigorously, we obtained high magnification confocal z-stacks of frozen thymic sections from CD11c-YFP chimeric mice in which ECs were stained with anti-CD31. This strategy allowed more accurate digital surface rendering of thymic DCs and the vascular lining to assess the morphology and size of intravascular protrusions of perivascular DCs (Fig. 3d and Supplementary Movie 4). Individual perivascular DCs were found to extend up to four processes toward the vessel lumen that were of variable size and irregular in shape. In some cases, the base of a protrusion formed a narrow neck

presumably due to constriction at the site where DCs penetrated the vessel wall, possibly poking through an endothelial junction (Fig. 3e, and Supplementary Movie 5). Although the interpretation of measurements of the apparent size of intra-and extra-vascular DC boundaries is limited by potential artifacts owing to tissue processing and the application of more or less-stringent post acquisition image rendering parameters, our analysis provides a reasonable estimate of the relative surface area of individual protrusions, which ranged from 3 μm² to 400 μm², corresponding to 0.2–25% of the DC's total surface area.

To further investigate this phenomenon in vivo, we administered IV injections of an anti-CD11c mAb conjugated to phycoerythrin (PE), a 240 kDa fluorescent protein with a mean diameter of 100 Å[21]. The large combined size of mAb-PE conjugates prevents them from quickly diffusing across the vascular wall, so after IV injection, these reagents bind initially only epitopes that are exposed to the blood circulation[22] (Fig. 4a). When mice were euthanized 2 minutes after anti-CD11c-PE injection, flow cytometry of thymic cell suspensions revealed that ~10% of DCs had bound the anti-CD11c mAb, but not an isotype-matched control mAb (Fig. 4b, c), suggesting that many perivascular DCs were at least partially exposed to the blood.

Several control experiments were performed to rigorously ascertain that anti-CD11c-PE specifically stained exposed intravascular epitopes on DCs. First, we surmised that the PE+ DC subset could have acquired unbound intravascular mAb during tissue dissociation for flow cytometry. To address this, we prepared single-cell suspensions after combining in the same dish thymi from two CD45 congenic mice, whereby one animal (CD45.1) had received anti-CD11c-PE IV, whereas the other (CD45.2) was left untreated. No fluorescent mAb was detectable on congenic DCs from uninjected donors (Supplementary Fig. 2a), indicating that PE+ DCs became labeled during the ~2 min window between IV mAb injection and tissue harvest. Second, we asked whether parenchymal DCs (par-DCs) could have acquired the PE label even though they were not directly exposed in the vessel lumen. For example, the thymic microvasculature is a site of constant bidirectional leukocyte trafficking, which could potentially result in occasional leaks that might enable access of blood-borne antibodies to extravascular cells in immediate proximity to such hypothetical leaks. To account for this alternative mechanism, we used the same intravascular labeling protocol to inject mice with anti-CD45-PE, which labels all accessible leukocytes. Blood and thymus were harvested 2 min. later and the frequency of CD11c+ DCs in the blood, among all CD45+ thymic leukocytes and among the PE+ thymic subset was assessed. As expected, CD11c+ DCs were rare among blood leukocytes (0.009% ± 0.001%) and among thymic mononuclear cells (0.3% ± 0.03%), while their frequency among PE+ thymic leukocytes (5.8% ± 0.63%) was more than one order of magnitude greater ($P < 0.0001$; Fig. 4d). This result is at odds with a vascular leak mechanism because anti-CD45-PE should have stained extravascular leukocytes indiscriminately resulting in similar DC frequencies among PE+ and total CD45+ thymic mononuclear cells. Similar results were obtained when we performed intravascular labeling using anti-H-2Kb-PE and evaluated PE+ cells among CD45+ hematopoietic and CD45− stromal cells. Specific staining was readily detectable in a fraction of hematopoietic cells, but there was no detectable signal above background on thymic stromal cells (Supplementary Fig. 2b), which should have been detected if the MAb had leaked into the surrounding tissue.

Next, we asked whether PE+ thymic DCs reflected cells that were located entirely in the vascular lumen (Lu-DCs) or represented transendothelial cells (TE-DCs) that resided partially in the extravascular space and partially in the bloodstream

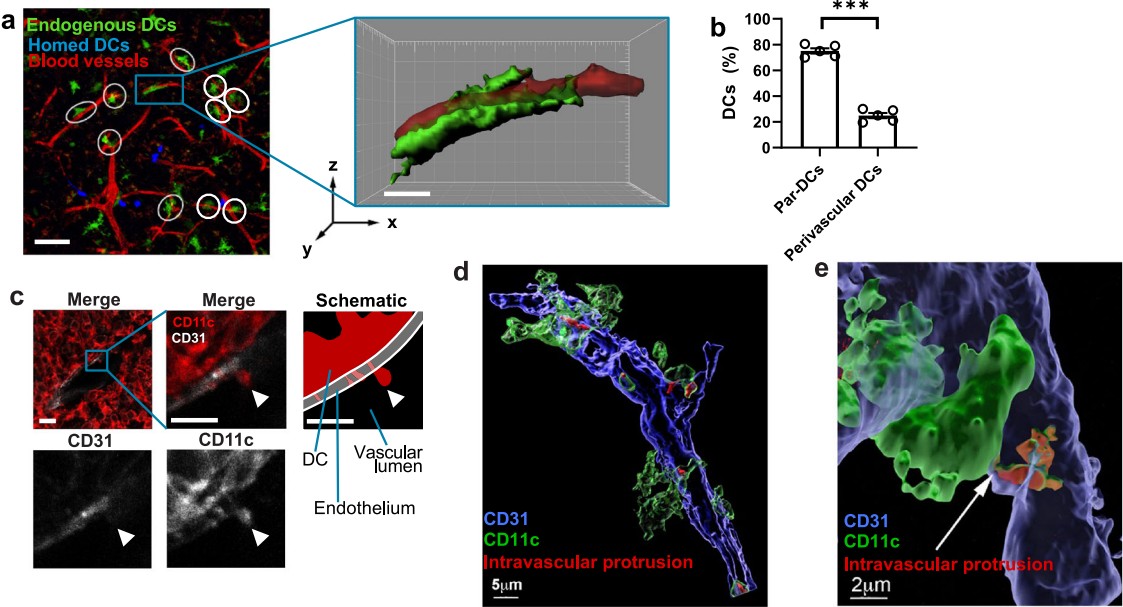

**Fig. 3 Identification of thymic TE-DCs. a** Representative MP-IVM micrograph of a thymus graft in a CD11c-YFP BM chimeric mouse (left panel). In all, 2 hours after adoptive transfer of labeled DCs, both homed DCs (blue) and endogenous DCs (green) are readily detectable in proximity to microvessels, which were visualized after injection of TRITC-labeled 2 MDa dextran (red). Circles identify typical DCs that are closely associated with microvessels. The box highlights one perivascular cell, which is shown in the right panel after the digital surface rendering of the DC and microvascular lumen (Supplementary Movie 3). Scale bars represent 50 μm (left panel) and 10 μm (middle panel), respectively. **b** The frequency of parenchymal and perivascular DCs in transplanted thymi was evaluated in confocal micrographs. Cells were considered as perivascular when the shortest distance between the margin of the cell and the lumen of the blood vessel was <1 μm. $n = 5$; ***$P = 0.0003$ (Student's two-tailed, paired $t$ test). **c** Confocal micrograph of a frozen thymic (ET) section showing CD11c+ DCs (red) extending protrusions through CD31+ vascular endothelium (white). The upper panels show merged two-color overlays at low (left) and high magnification (middle) and a schematic illustration of the latter (right). The lower panels show single channels for CD31 (left) and CD11c staining (right) of the same high-power field. Scale bars reflect 10 μm. **d** 3D reconstruction and surface rendering from a confocal z-stack of a frozen thymic section similar to that described in **c**, showing the vasculature in blue (CD31 staining), perivascular CD11c+ DCs in green (GFP signal), and their intravascular protrusions pseudocolored in red. To facilitate visualization of the transendothelial protrusions, only the perivascular GFP+ DCs are shown in the analysis. (Supplementary Movie 4). **e** Zoomed-in detail showing the constriction of the cell body of a perivascular GFP+ DC presumably protruding across the endothelial junction (arrow). *DCs* dendritic cells, *Par-DCs* parenchymal dendritic cells. See also Supplementary Movie 5.

(Fig. 4a), as suggested by our observations in frozen thymic section (Fig. 3c–e). To discriminate between these two (non-exclusive) alternatives, we devised a competitive staining protocol whereby mice were either injected with anti-CD11c-PE (clone N418) or PE-labeled isotype control mAb. Animals were euthanized 2 min. later and single-cell suspensions were prepared from thymus, spleen, blood, and peripheral lymph nodes (LNs) and stained ex vivo with anti-CD11c-PE-Cy7 (clone N418). This protocol ensured that all cell surface CD11c molecules were bound by mAb; those exposed to mAb in vivo were tagged with PE (peak emission 575 nm), whereas epitopes were inaccessible to circulating mAb were identified after ex vivo labeling with PE-Cy7 (peak emission 767 nm). In addition, as a measure of the total copy number of CD11c epitopes per cell, DCs were stained with an AF647-labeled mAb (peak emission 660 nm) against hamster IgG, which recognizes mAb N418 irrespective of the associated fluorophore (Supplementary Fig. 3). This approach allowed us to determine for each cell in vivo and ex vivo labeling indices, which provided a quantitative measure of the fraction of epitopes that were accessible for staining in vivo (i.e., ratio of fluorescence intensities (FI) at 575 nm ($FI_{575}$ to $FI_{660}$) and after cell isolation ex vivo ($FI_{767}$: $FI_{600}$), respectively. Results were visualized as two-dimensional dot plots, which allowed gating on specific DC subsets.

To set a gate encompassing Par-DCs, which by definition should not bind intravascular mAb, we performed the above procedure in animals that had been injected with PE-labeled isotype control mAb (Fig. 4e). As expected, DCs in this sample

had a uniformly high ex vivo labeling index and low in vivo labeling index. Conversely, in animals injected with anti-CD11c-PE peripheral blood DCs displayed a high in vivo labeling index and low ex vivo labeling index, as expected for Lu-DC. To meet the definition of TE-DC, in which only a fraction of CD11c epitopes is accessible from the blood, we defined a third gate in which cells display greater ex vivo and in vivo labeling indices than Lu-DCs and Par-DCs, respectively. Such cells were readily detectable among thymic DCs where they represented ~10% of the overall DC pool. Essentially all remaining CD11c+ cells were Par-DCs, while Lu-DCs were absent. By contrast, all three DC populations were readily detectable among splenic DCs, whereas LN DCs were exclusively composed of Par-DCs (Fig. 4e, f). These findings demonstrate that ~10% of thymic DCs bridge between the intra- and extravascular thymic compartment, presumably by extending processes across the endothelial barrier, which allows detection by blood-borne antibodies.

**Thymic TE-DCs are phenotypically distinct from Par-DCs.** Next, we asked whether TE-DCs are representative of the entire pool of thymic DCs or denote a phenotypically distinct subset. A fluorescence-activated cell sorting (FACS) analysis of DC subset markers revealed that thymic Par-DCs closely reflect the overall DC subset composition in ET and TT (Fig. 1e), whereas thymic TE-DCs were markedly distinct and closely resembled the DC composition in spleen and peripheral LNs (Fig. 4g). Specifically, thymic TE-DCs contained a much larger fraction of

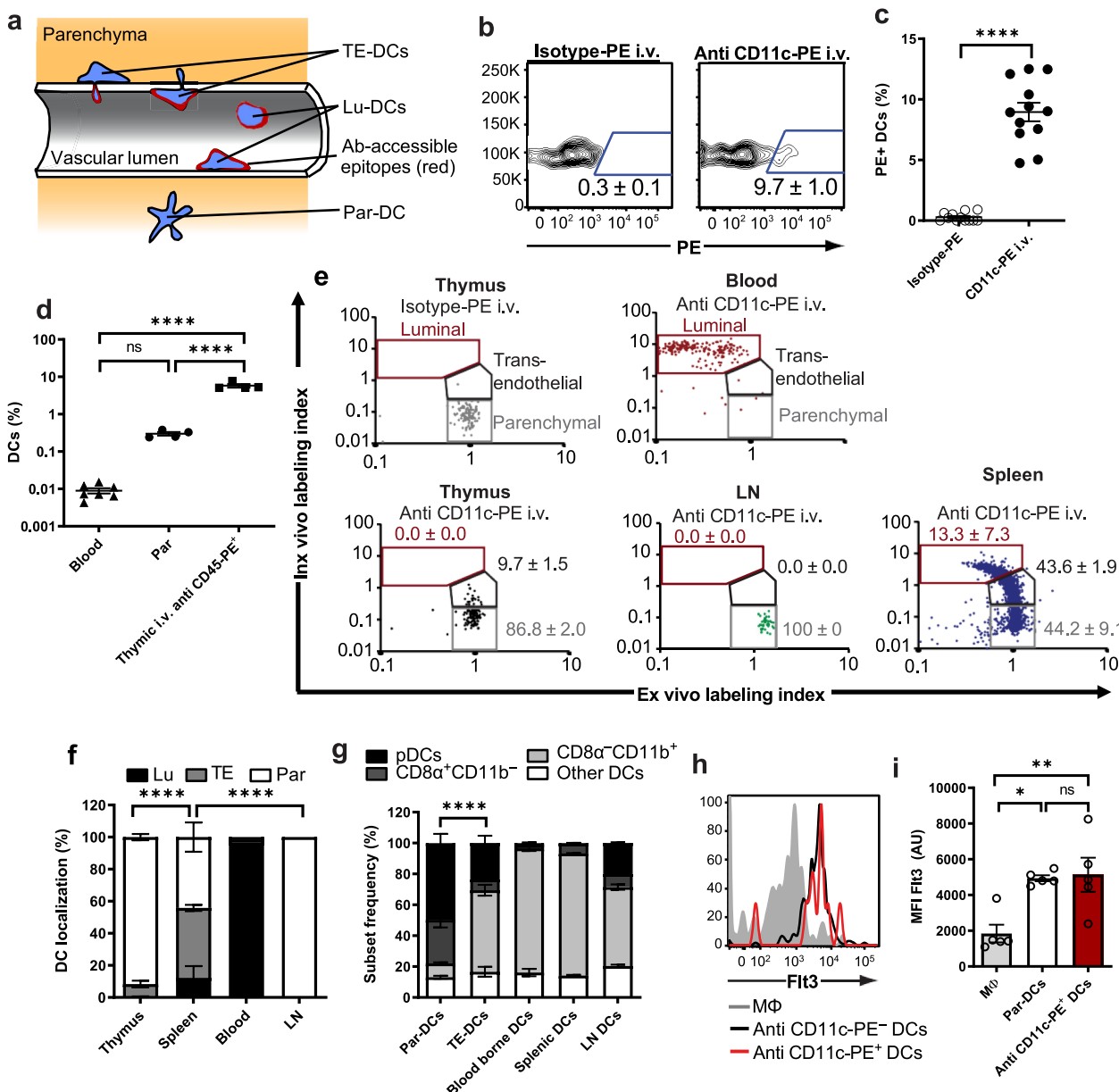

**Fig. 4 Characterization of thymic TE-DCs. a** Schematic illustration of the distribution of thymic DCs in the parenchyma (Par-DC), vessel lumen (Lu-DC), and transendothelial localization (TE-DC) and the expected accessibility of epitopes (red) for IV injected mAbs. **b** and **c** C57BL/6 mice were injected IV with anti-CD11c-PE or isotype-PE mAb and euthanized 2 minutes later. **b** Representative FACS plots showing PE+ DCs in ET single-cell suspensions after gating on total CD11c+ DCs. **c** Frequency of thymic PE+ DCs from two independent experiments; $n = 12$ mice/group; $n = 5$; ****: $P < 0.0001$ (Student's two-tailed, unpaired $t$ test). **d** Mice were injected with anti-CD45-PE mAb and euthanized after 2 mins. The frequency of DCs was determined among total blood and thymic leukocytes and among CD45-PE+ thymic leukocytes. $n = 5$ mice (one-way ANOVA, n.s.: not significant, ****$P < 0.0001$). **e** Representative plots of in vivo and ex vivo labeling indices of DCs from blood, thymus, lymph nodes (LN), and spleen that were determined as described in Results and Supplementary Fig. 3. Gates reflecting parenchymal and luminal localization were set based on labeling indices obtained from thymic DCs after injection of isotype-PE and analysis of anti-CD11c-PE stained blood-borne DCs, respectively. For a transendothelial localization, in which only a fraction of CD11c epitopes should be accessible from the blood, a third gate was defined in which cells display greater ex vivo and in vivo labeling indices than luminal and parenchymal DCs, respectively. **f** Percentage of Lu-DC, Par-DC, and TE-DCs in thymus, spleen, blood, and LN. $n = 4$ mice/group (two-way ANOVA, ****$P < 0.0001$). **g** DC subset composition of thymic Par-DC and TE-DCs as well as blood-borne, splenic, and LN DCs. $n = 5$ mice/group (two-way ANOVA, ****$P < 0.0001$). **h** and **i** Thymic macrophages (MΦ), Par-DCs, and PE+ DCs were evaluated for Flt3 expression. **h** Representative FACS histogram; **i** mean fluorescence intensities (MFI) of each cell population. $n = 3$ independent experiments with five mice/each (one-way ANOVA, **$P = 0.0078$, *$P = 0.0115$). Bars and error bars represent mean ± SEM. See also Supplementary Figs. 2–4.

$CD8\alpha^-CD11b^+$ cells and fewer pDCs and $CD8\alpha^+CD11b^-$ cells than thymic Par-DCs.

Since the above experiments relied on CD11c as a marker for DCs, it was important to rule out that our analysis of TE-DCs included other leukocytes, such as macrophages, which can express CD11c in some tissues[23]. TE-DCs and bulk thymic $CD11c^+$ cells uniformly expressed identical levels of Flt3, c-Kit, and CD26, a phenotype that is characteristic of bona fide DCs[23]. By contrast, recent single-cell transcriptomic studies of thymic leukocytes indicate that Flt3 is not expressed on macrophages, pDCs, or a recently identified monocyte-derived thymic DC population (Fig. 4h, i and Supplementary Fig. 4)[24].

**Origin of thymic TE-DCs.** Having established the identity of thymic TE-DCs, we next sought to determine the origin of this unusual DC population. As TE-DCs were enriched in the $CD8\alpha^-CD11b^+$ subset (Fig. 4g), which is prominent in peripheral blood, but rare among parenchymal thymic DCs (Fig. 1e), we asked whether TE-DCs originate in the periphery. To this end, we adoptively transferred $1 \times 10^7$ congenic splenic DCs, which began to accumulate in the thymus within 2 h and reached peak numbers at 18 h post injection (Supplementary Fig. 5), consistent with previous findings[16]. This was paralleled by a doubling in the frequency of thymic DCs that bound IV injected anti-CD11c-PE (Fig. 5a). This temporary rise in PE$^+$ DCs could be accounted for, in part, by the transient appearance of Lu-DCs (Fig. 5b), but was primarily caused by a marked increase in TE-DCs (Fig. 5c). These changes did not affect ET DCs whose frequency and localization remained constant (Fig. 5d). Rather, the transient increase in PE$^+$ DCs apparently reflected the homing of transferred congenic DCs, which were almost exclusively luminal at 2 h post transfer, contributed to the pool of TE-DCs and Par-DCs between 18 h and 48 h, and by day 7 after the adoptive transfer had assumed an entirely extravascular localization (Fig. 5e).

As a second approach to address the origin of thymic TE-DCs, we performed parabiosis experiments whereby pairs of congenic mice were surgically joined to establish a shared blood circulation (Fig. 5f). Two weeks after surgery, the chimerism (i.e., the percentage of partner-derived cells in each parabiont) of blood lymphocytes, blood DCs and thymic DCs was $41.5 \pm 1.7\%$, $33.3 \pm 4.3\%$, and $9.4 \pm 1.3\%$, respectively. By contrast, thymocyte chimerism was very low ($0.5 \pm 0.3\%$), consistent with the finding that at this time point partner-derived Lin$^-$c-kit$^+$CD25$^+$ early thymic progenitors (ETPs) were undetectable in recipient thymi. Since ETPs are thought to include the precursors that give rise to the ET DC population[25–27], it is unlikely that the partner-derived thymic DCs in parabiotic mice arose from intrathymic progenitors. Presumably, these cells entered the thymus as fully differentiated DCs from the blood[16]. Of note, upon IV injection of anti-CD11c-PE into parabiotic animals, thymic DC chimerism was much greater among PE$^+$ TE-DCs than among PE$^{neg}$ Par-DCs ($29.4 \pm 6.8\%$ vs. $9 \pm 2\%$, respectively, Fig. 5f, g). Partner-derived thymic DCs were also significantly enriched in $CD11c^+CD8\alpha^-CD11b^+$ DCs with a correspondingly lower frequency of $CD11c^+CD8\alpha^+CD11b^-$ DCs than among ET DCs (Fig. 5h).

These findings raised the question of whether TE-DCs that are recruited from the blood remain stationary in their unique transcompartmental position or whether they represent motile cells that bind anti-CD11c-PE while diapedesing from the intra- to the extravascular compartment. If the latter was the case, TE-DCs would need to be constantly and rapidly replenished by newly homed cells from the circulating DC pool. As shown above (Fig. 1h) and previously reported[16], homing of blood-borne DCs to the thymus is mediated by a multi-step adhesion cascade that

critically depends on $\alpha_4$ integrins. Thus, we pretreated mice with anti-$\alpha_4$ integrin mAb 18 h before IV injection of anti-CD11c-PE. Remarkably, inhibition of DC homing minimally affected the frequency of PE$^+$ DCs (Fig. 5i). Even after continuous blockade of $\alpha_4$ integrins for 7 days the frequency of TE-DCs was only reduced by ~half. Moreover, prolonged inhibition of $\alpha_4$ integrins did not alter the subset composition of Par-DCs, whereas among PE$^+$ DCs the $CD11c^+CD8\alpha^-CD11b^+$ subset was progressively lost (Fig. 5j). This indicates that TE-DCs are not merely DCs "in transit", i.e., in the process of homing into the thymic parenchyma. Rather, our results suggest that most TE-DCs, particularly the $CD8\alpha^-CD11b^+$ subset, are derived from fully differentiated circulating DCs, which adhere to thymic micro-vessels and then migrate partially across the vessel wall to establish a sustained presence at the thymus-blood interface. The apparent transendothelial dwell time of this TE-DC population lasts at least 48 h and possibly longer than a week. In addition, it should be noted that our findings do not rule out that some TE-DCs (especially the minor $CD8\alpha^+CD11b^-$ subset and perhaps also some pDCs) could arise also from intrathymic progenitors with potentially distinct turnover rates.

**Localization of thymic TE-DCs depends on the $CX_3CR1$–$CX_3CL1$ pathway.** Next, we set out to identify the thymic guidance signals for transendothelial positioning of DCs. TE-DCs are primarily composed of $CD11c^+CD8\alpha^-CD11b^+$ cells (Fig. 4g), which are known to express the chemokine receptor $CX_3CR1$, whereas other DC subsets express little or no $CX_3CR1$ (Supplementary Fig. 6a). Accordingly, in Transwell chemotaxis assays $CD8\alpha^-CD11b^+$ DCs as well as pDCs, but not $CD8\alpha^+CD11b^-$ DCs, migrated toward $CX_3CL1$ in a dose-dependent manner (Fig. 6a, b). Thus, we examined thymic DCs in mice deficient in $CX_3CR1$ or its ligand, $CX_3CL1$. Indeed, in the absence of either member of this chemokine pathway the frequency of DCs that were stained by intravascular anti-CD11c-PE was markedly reduced in the thymus, but not in blood or spleen (Fig. 6c and Supplementary Fig. 6b, c). Consequently, although the frequency of thymic TE-DCs in mutant mice was reduced by two- to threefold as compared with wild-type thymi, the fraction of par-DCs was proportionally increased (Fig. 6d, e). This change was not a consequence of any gross alteration of thymic DCs, as the overall number and composition of DC subsets were indis-tinguishable between wild-type and $Cx3cr1^{gfp/gfp}$ or $Cx3cl1^{-/-}$ animals (Supplementary Fig. 6d, e). $CX_3CR1$ also appears to be dispensable for DC homing to the thymus; neither adoptive transfers (Supplementary Fig. 6f) nor parabiosis between wild-type and $Cx3cr1^{gfp/gfp}$ partners (Fig. 6f) revealed any defect in DC trafficking to the thymus. Of note, in parabiotic pairs of WT and $Cx3cr1^{gfp/gfp}$ mice the frequency of thymic TE-DCs was equiva-lent, indicating that DCs from the wild-type partner had repo-pulated the vacant transendothelial niches in thymi of their Cx3cr1-deficient counterparts (Fig. 6g).

Since the above findings implied that $CX_3CL1$ should be directly associated with thymic microvessels, we examined $CX_3CL1^{mCherry}$ reporter mice (Supplementary Fig. 7). The spectral properties of the mCherry fluorescent protein were not conducive to visualization by MP-IVM, but the $CX_3CL1$ reporter was readily detectable in frozen thymic sections by confocal microscopy. Interestingly, simultaneous immunostaining for CD31 and ACKR1, markers that identify ECs in all microvessels or exclusively in postcapillary venules, respectively[28], revealed that mCherry and ACKR1 expression was mutually exclusive. Thus, $CX_3CL1$ was preferentially expressed in non-venular segments of the microvascular network, particularly capillaries and small arterioles.

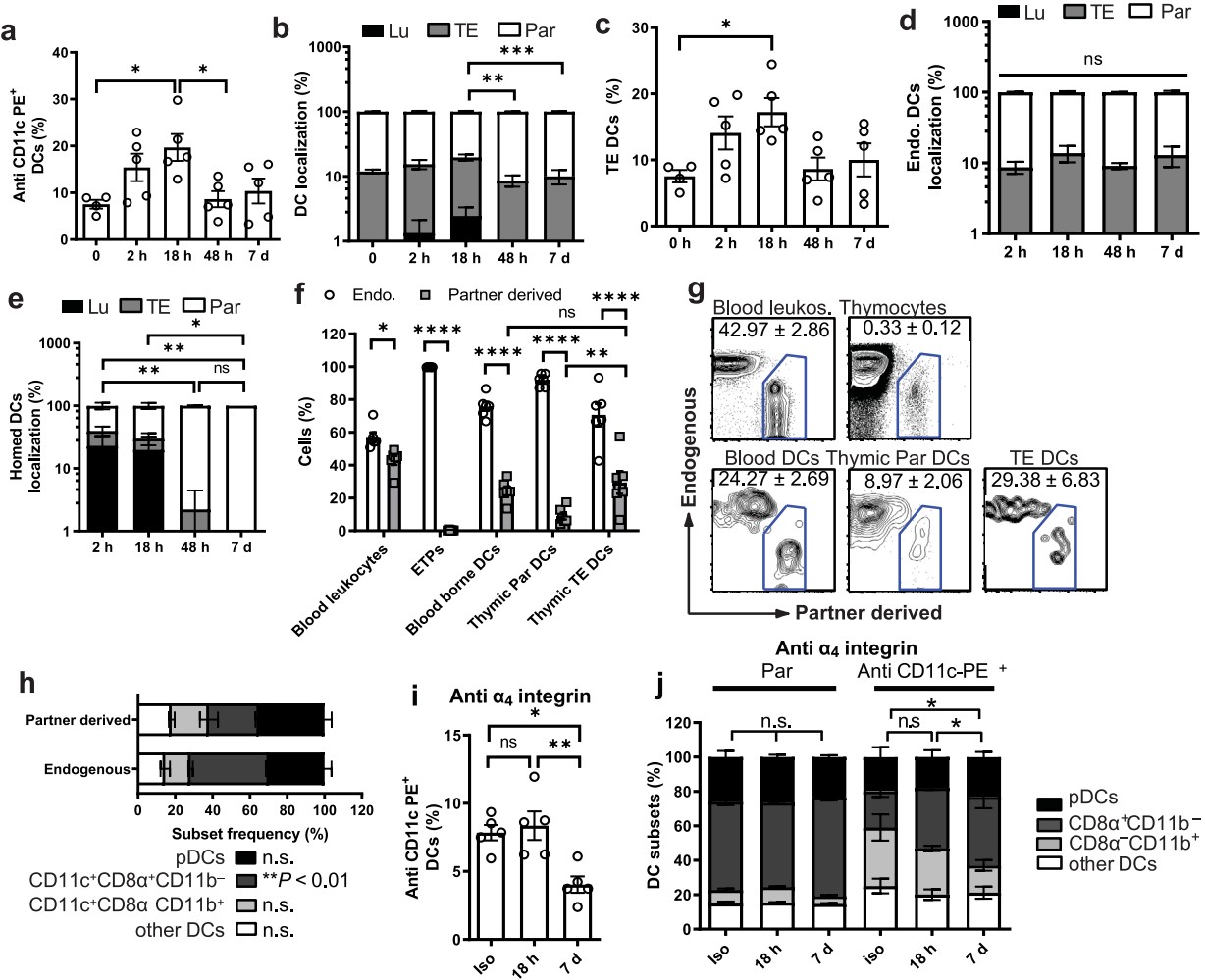

**Fig. 5 Origin of thymic TE-DCs. a–e** C57BL6 mice (CD45.1[+]) were injected with congenic (CD45.2[+]) splenic DCs and analyzed at indicated time points. Two minutes prior to tissue harvest, mice received anti-CD11c-PE mAb IV ($n = 3$ experiments with five mice/group). Labeling indices were determined to identify DC localization as illustrated in Supplementary Fig. 3 and Fig. 4a. **a** Frequency of PE[+] DCs among total DCs recovered from thymi before and after adoptive DC transfer (one-way ANOVA). **b** Frequency of TE-DC, Lu-DC, and Par-DC among total thymic DCs (two-way ANOVA) and **c** frequency of TE-DCs at indicated time points before and after DC transfer (one-way ANOVA). **d** Localization of endogenous DCs and **e** homed DCs at the indicated time points (two-way ANOVA). **f–h** Parabiotic pairs were generated by surgically joining age- and sex-matched CD45.1[+] and CD45.2[+] congenic mice. Two weeks after parabiosis surgery, animals were injected with anti-CD11c-PE and tissues were harvested 2 mins later. $n = 2$ experiments with five mice/group (two-way ANOVA). **f** Frequency of endogenous (empty bars) and partner-derived (filled bars) leukocytes, early thymic progenitors (ETPs), and DCs in blood and thymus. **g** Representative FACS plots of leukocytes and DCs in parabiotic animals. **h** Subset composition of endogenous and partner-derived thymic DCs. **i** and **j** Effect of α4 integrin inhibition on thymic DC subsets. C57Bl/6 mice were either treated for up to 7 days with a blocking anti-α4 integrin mAb or isotype control (Iso). Mice received anti-CD11c-PE mAb 2 mins prior to tissue harvest ($n = 2$ experiments with five mice/group; two-way ANOVA). **i** Frequency of CD11c-PE[+] cells among total thymic DCs (one-way ANOVA). **j** Subset composition of thymic Par-DC and anti-CD11c-PE[+] DCs (two-way ANOVA). Bars and error bars represent mean ± SEM. **a** *$P = 0.0236$ (0 h vs. 18 h) and $P = 0.0302$ (18 h vs. 48 h). **b** ***$P = 0.0009$, **$P = 0.0068$. **c** *$P = 0.0405$. **e** **$P = 0.005$ (2 h vs. 48 h) and $P = 0.005$ (2 h vs. 7d), *$P = 0.0488$ (18 h vs. 7d). **f** **$P = 0.0019$, *$P = 0.041$, ****$P < 0.0001$. **i** **$P = 0.005$, *$P = 0.0115$. **j** *$P < 0.05$; n.s. not significant, Lu luminal dendritic cells, TE transendothelial dendritic cells, Par parenchymal denritic cells, pDCs plasmacytoid dendritic cells. See also Supplementary Fig. 5.

**TE-DCs capture and present blood-borne antigens for negative selection**. In light of the fact that TE-DCs have simultaneous access to both the intravascular space and the extravascular thymic parenchyma, we hypothesized that TE-DCs might capture and process circulating Ag for presentation to nearby extravascular thymocytes. To test this hypothesis, we injected fluorescent chicken ovalbumin (OVA-AF647) IV to assess whether thymic DCs can acquire blood-borne proteins. OVA is a 45 kDa molecule with a spherical diameter of ~50 Å[29], dimensions that render the intact protein too large to passively diffuse across the blood–thymus barrier[30]. As early as 5 mins after IV injection of OVA-AF647, a fraction of thymic DCs had acquired the fluorescent probe in a dose-dependent fashion (Fig. 7a). Upon co-injection of OVA-AF647 with anti-CD11c-PE, as expected, the vast majority of thymic DC were PE[neg] Par-DCs, of which only very few contained detectable (and mostly very low) amounts of OVA-AF647, whereas 38% of PE[+] TE-DCs were OVA-AF647[+] (Fig. 7b, c). It is likely that OVA uptake by DCs was lagging behind intravascular anti-CD11c-PE labeling, presumably owing to the much faster kinetics of MAb binding to luminal epitopes as

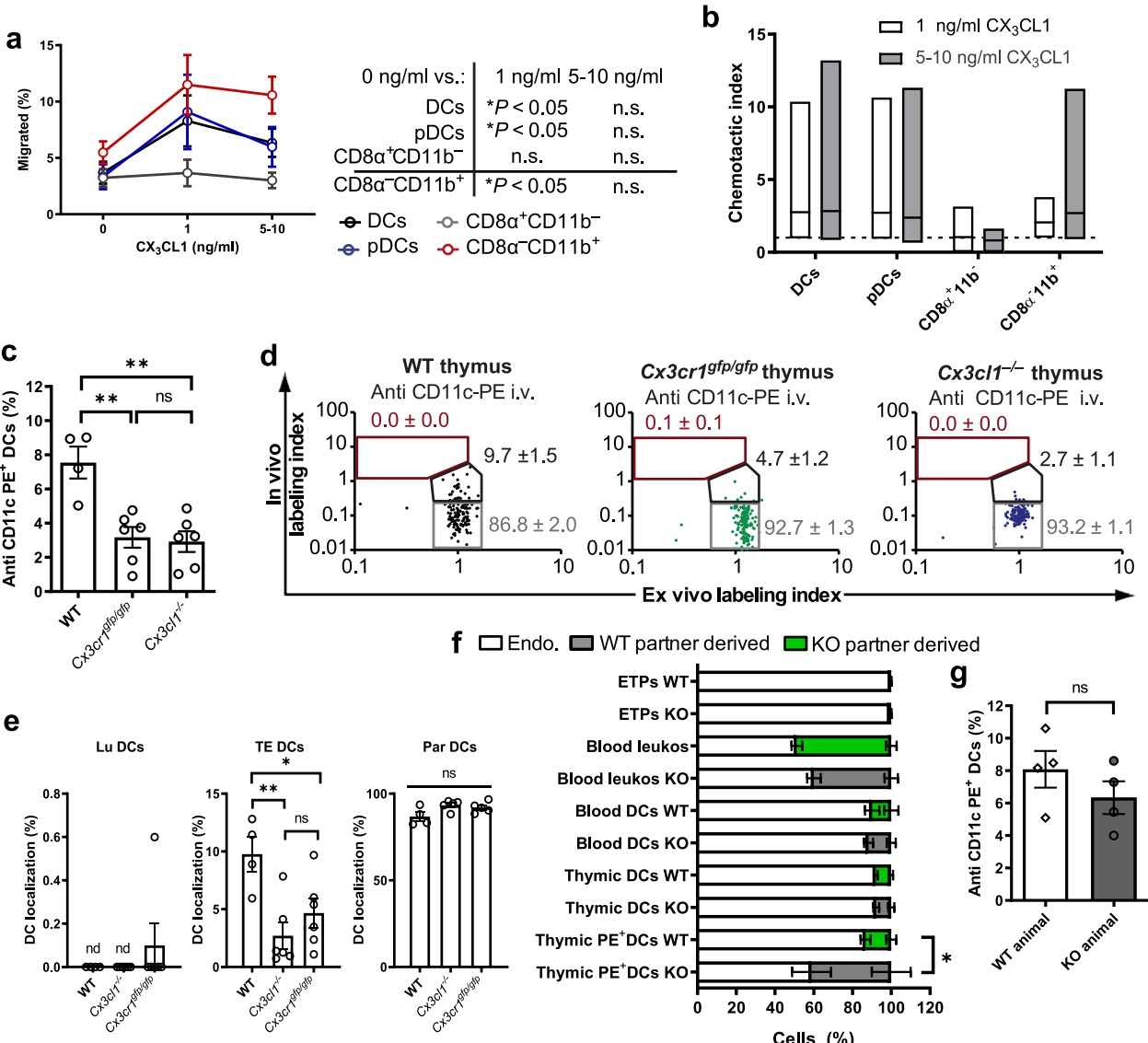

**Fig. 6 Role of the CX₃CR1-CX₃CL1 pathway in thymic TE-DC positioning.** Thymic DC migration to gradients of CX₃CL1 was assessed in transwell chemotaxis assays. Data are presented as **a** the percentage of migrated cells in each DC subset and **b** as a chemotactic index (ratio of the number of cells that migrated to media alone vs. media containing CX₃CL1). $n = 3$ independent experiments with DCs from three to nine mice per group, two-way ANOVA. **c–e** Age and sex-matched C57B6 (WT), Cx3cr1$^{gfp/gfp}$, and Cx3cl1$^{-/-}$ mice were injected with anti-CD11c-PE mAb IV and the thymus was harvested 2 min later to analyze **c** the frequency of PE⁺ DCs by FACS. $n = 3$ experiments, each with five mice/group (one-way ANOVA) and **d** intrathymic DC localization in each individual animal (representative examples are shown for each genotype) and **e** at a population level. $n = 2$ independent experiments, five mice/group (two-way ANOVA). **f** Leukocyte chimerism in blood and thymi of parabiotic Cx3cr1$^{gfp/gfp}$ (CD45.1⁺) and C57B6 (WT; CD45.2⁺) parabiotic mice. $n = 7$ pairs of mice (two-way ANOVA). **g** Parabiotic mice were injected with anti-CD11c-PE, euthanized after 2 min. and the frequency of PE⁺ cells among total thymic DCs was assessed in each partner. $n = 5$ pairs of mice (unpaired, two-tailed Student's $t$ test). Bars and error bars represent mean ± SEM. **c** **$P = 0.0024$ (WT vs. Cx3cr1$^{gfp/gfp}$) and $P = 0.0015$ (WT vs. Cx3cl1$^{-/-}$), n.s.: not significant. **e** **$P = 0.0071$, *$P = 0.0475$. **f** *$P = 0.0356$. *DCs* dendritic cells, *pDCs* plasmacytoid dendritic cells, *Lu* luminal dendritic cells, *TE* transendothelial dendritic cells, *Par* parenchymal dendritic cells. See also Supplementary Figs. 6 and 7.

compared with endocytic OVA uptake by TE-DCs. However, we cannot rule out that at least some TE-DCs were incapable of acquiring plasma proteins. Notwithstanding, the DC population that had acquired OVA does not appear to reflect a discrete subset because, like bulk TE-DCs, the OVA-AF647⁺ TE-DC subset preferentially displayed a CD8α⁻CD11b⁺ phenotype (Fig. 7d). Importantly, the frequency of OVA-AF647⁺ DCs among total thymic DCs was markedly reduced in Cx3cr1$^{gfp/gfp}$ and Cx3cl1$^{-/-}$ mice (Fig. 7e), consistent with the profound paucity of TE-DCs in those animals (Fig. 6c).

Having determined that thymic TE-DCs can rapidly acquire circulating plasma proteins, we tested whether TE-DCs process and present the acquired material to thymocytes. As DCs are adept at cross-presenting exogenous Ag in major histocompatibility complex (MHC) class I[31], we devised experiments that rely on this cross-presentation capacity. To this end, groups of wildtype, Cx3cr1$^{gfp/gfp}$, and Cx3cl1$^{-/-}$ mice received 100 μg unlabeled OVA intravenously. Animals were killed 5 minutes later and thymic DCs were isolated and tested in vitro for their ability to delete CD4⁺CD8⁺ thymocytes from OT-I mice, which

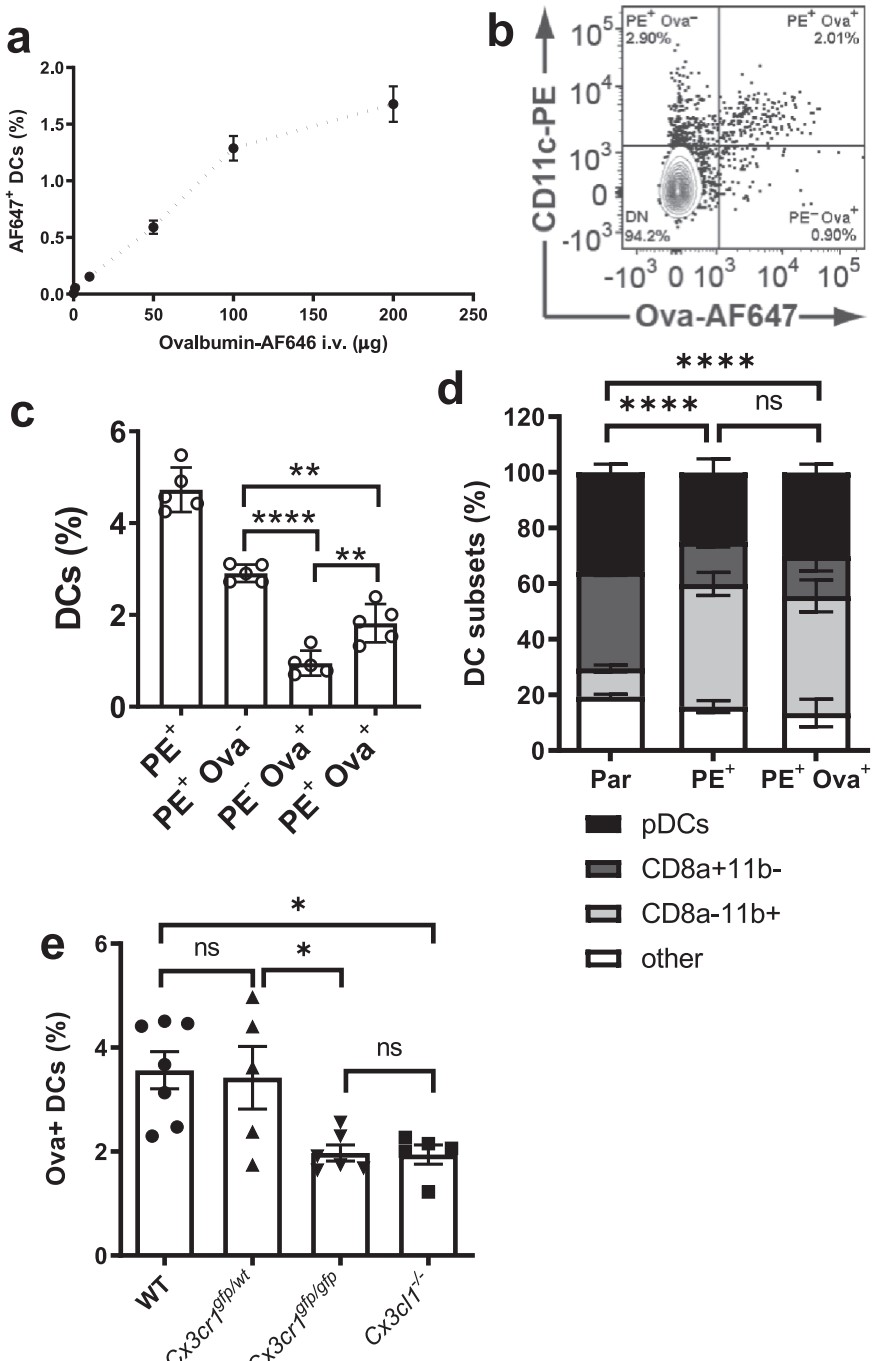

**Fig. 7 Transendothelial thymic DCs capture blood-borne proteins. a** Uptake of Alexa 647-labeled ovalbumin (Ova-AF647) by thymic DCs. Different doses of Ova-AF647 were injected IV and thymic DCs were harvested 5 min later. $n = 3$ experiments with three mice/group for 0 and 1 μg, four mouse/ group for 10, 50 100, and 200 μg. **b** Representative FACS plot of thymic DCs. In all, 5 min after IV co-injection of anti-CD11c-PE and 100 μg Ova-AF647, thymic single-cell suspensions were obtained and stained for CD11c$^+$ DC. OVA-AF647 geometric mean fluorescence intensity (geoMFI) for PE$^+$ subset (TE-DC) = 982 and for PE$^{neg}$ subset (Par-DC) = 75.1 (a.u.). **c** Frequency of thymic DCs that bound IV injected anti-CD11c-PE (PE$^+$) and/or Ova-AF647. $n = 2$ independent experiments with five mice/group each, of which one representative experiment is shown (one-way ANOVA). **d** Subset analysis of thymic Par-DC (i.e., PE$^{neg}$), PE$^+$ DC (i.e., TE-DC) that did or did not acquire Ova-AF647. $n = 2$ independent experiments with 10 mice/group (two-way ANOVA). **e** Age and sex-matched wildtype (WT), Cx3cr1$^{gfp/+}$, Cx3cr1$^{gfp/gfp}$, and Cx3cl1$^{-/-}$ mice were injected with 100 μg Ova-AF647 and thymi were harvested 5 min later for FACS analysis of DCs. $n = 2$ experiments with five mice per group (one-way ANOVA). Bars and error bars represent mean ± SEM. **c** ****$P < 0.0001$; **$P = 0.0011$ (PE$^+$Ova$^-$ vs. PE$^+$Ova$^+$) and $P = 0.0073$ (PE$^-$Ova$^+$ vs. PE$^+$Ova$^+$). **d** ****$P < 0.0001$. **e** *$P = 0.0217$ (WT vs. Cx3cl1$^{-/-}$) and $P = 0.0172$ (WT vs. Cx3cr1$^{gfp/gfp}$), *DCs* dendritic cells, *pDCs* plasmacytoid dendritic cells.

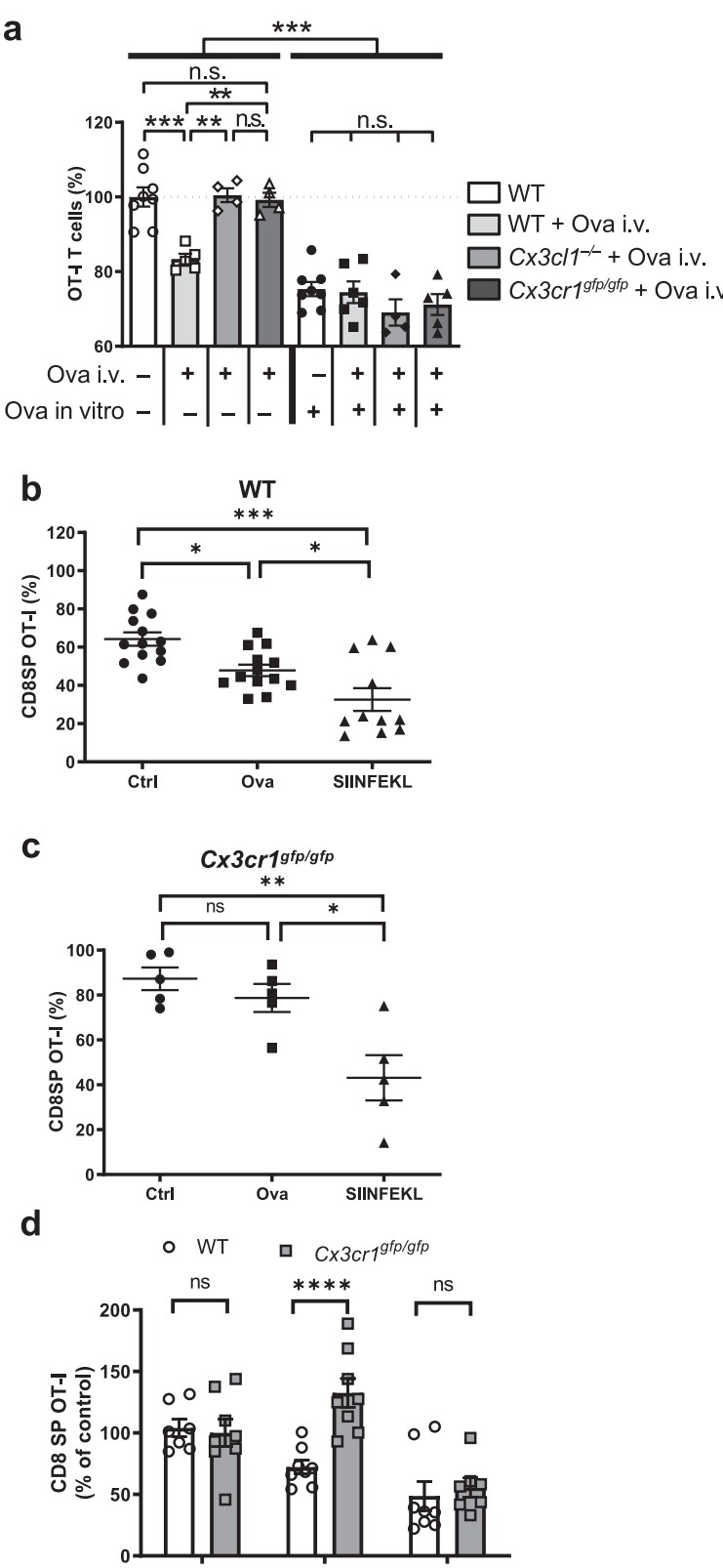

express a transgenic TCR specific for the OVA-derived SIINFEKL peptide in H2-k$^b$ [32]. Indeed, thymic DCs from OVA injected wild-type mice induced statistically significant thymocyte deletion when compared with thymic DCs of OVA-naive mice (P < 0.001; Fig. 8a). By contrast, thymic DCs from OVA-treated Cx3cr1$^{gfp/gfp}$ and Cx3cl1$^{-/-}$ mice were incapable of deleting OT-I thymocytes. This difference was a consequence of compromised in vivo access

of CX$_3$CR1-deficient DCs to Ag and not owing to a cell-intrinsic defect in Ag uptake, processing or presentation because in vitro addition of OVA to DC-thymocyte co-cultures resulted in equivalent deletion regardless of DC genotype.

To address whether TE-DCs could also present cognate Ag to developing thymocytes in vivo, we reconstituted lethally irradiated C57BL/6 mice (CD45.1) with mixed BM from OT-I

**Fig. 8 TE-DCs mediate negative selection against blood-borne antigens. a** Purified thymic dendritic cells (DCs) from untreated or ovalbumin (Ova) injected wildtype, Cx3cr1$^{gfp/gfp}$ or Cx3cl1$^{-/-}$ mice were co-cultured for 3 days with CD4$^+$CD8$^+$ OT-I thymocytes with or without adding exogenous Ova. The number of viable OT-I cells in each sample was assessed by flow cytometry and compared to control co-cultures of WT DCs without Ova. $n = 2$ experiments with DC preparations from 4–8 mice/group ($n = 4$ for Cx3cl1$^{-/-}$ + Ova i.v., Cx3cr1$^{gfp/gfp}$+ Ova i.v. and Cx3cl1$^{-/-}$ + Ova i.v. + Ova in vitro; $n = 5$ for WT + Ova i.v., Cx3cr1$^{gfp/gfp}$ + Ova i.v. + Ova in vitro; $n = 6$ for WT + Ova i.v. + Ova in vitro; $n = 8$ for WT + PBS i.v. and WT + PBS i.v. + Ova in vitro; one-way ANOVA). **b** Lethally irradiated CD45.1 or (**c**) CX$_3$CR1$^{-/-}$ × CD45.1 animals were reconstituted with mixed BM at a 1:1 ratio of **b** CD45.2 and OT-I × CD45.1/2 or **c** CD45.1 × CX$_3$CR1$^{-/-}$ and OT-I × CD45.1/2 × CX$_3$CR1$^{-/-}$ BM. Deletion of OT-I T cells was induced 18 days after bone marrow transplantation by injection of 100 μg ovalbumin + 100 μg non-depleting anti-α$_4$ integrin mAb (Ova) or 10 μg SIINFEKL peptide + 100 μg non-depleting anti-α$_4$ integrin mAb (SIINFEKL). Control mice were left untreated (Ctrl). Deletion was assessed by determining the percentage of CD8$^+$ OT-I cells among all CD8$^+$ cells in each animal by flow cytometry. Pooled data with $n = 13$ (Ctrl and Ova) and $n = 11$ (SIINFEKL) of two independent experiments is shown (**b**) or 5 animals (**c**) per group (one-way ANOVA). **d** Presentation of data in **b** and **c** as the percentage of CD8$^+$ OT-I T cells relative to control animals in WT (open bars) and Cx3cr1$^{gfp/gfp}$ (closed bars) animals. $n = 8$ for all groups except $n = 7$ for WT ctrl and Cx3cr1$^{gfp/gfp}$ SIINFEKL. Bars and error bars represent mean ± SEM. **a** ***$P = 0.005$, **$P = 0.0027$ for WT Ova i.v vs Cx3cl1$^{-/-}$ Ova i.v. and **$P = 0.0068$ for WT Ova i.v vs Cx3cr1$^{gfp/gfp}$ Ova i.v.; n.s.: not significant. **b** *$P = 0.0183$ (Ctrl vs Ova) and $P = 0.0384$ (Ova. vs. SIINFEKL), ****$P < 0.0001$. **c** **$P = 0.0033$, *$P = 0.0141$, n.s. not significant. **d** ***$P = 0.0002$.

donors expressing CD45.1/2 and congenic wild-type mice expressing CD45.2 at a 1:1 ratio. Eighteen days later when graft-derived thymocytes had repopulated the thymus, but fully mature T cells had not yet egressed to the periphery, animals were injected with anti-α$_4$ integrin mAb, and immediately thereafter mice received 100 μg full-length OVA or SIINFEKL peptide or were left untreated. Deletion of OVA-specific CD45.1$^+$ OT-I thymocytes was assessed 24 h later. Concomitant treatment with anti-α$_4$ integrin mAb was indispensable for these experiments to prevent Ag delivery by thymus-bound circulating DCs. In wild-type recipients, injection of both SIINFEKL as well as full-length OVA resulted in the deletion of OT-I cells, although the latter was somewhat less efficient (Fig. 8b).

To test whether this effect required the action of TE-DCs, the experiment was then repeated in lethally irradiated Cx3cr1$^{GFP/GFP}$ mice that were transplanted with mixed BM from Cx3cr1$^{GFP/GFP}$ and Cx3cr1$^{GFP/GFP}$ × OT-I mice at a 1:1 ratio. Again, injection of SIINFEKL peptide resulted in the deletion of OT-I thymocytes, whereas full-length OVA failed to do so (Fig. 8c). In fact, there was no difference in deletion efficiency between wild-type and CX$_3$CR1-deficient recipients of SIINFEKL peptide, whereas there was a significant difference between the two strains ($P < 0.05$) after full-length OVA injection (Fig. 8d). We conclude that a specialized subset of thymic DCs relies on CX$_3$CR1 to assume a transendothelial position that allows these cells to acquire blood-borne antigenic macromolecules and to make the acquired material available for tolerance induction in developing T cells.

**Identification of TE-DCs in human thymi.** Having established the presence and function of TE-DCs in murine thymi, we asked whether thymic TE-DCs occur also in other species. To this end, we performed immunofluorescence microscopy in humanized BLT mice, which had been generated by grafting human fetal thymi and liver under both kidney capsules and concomitantly transplanting autologous human fetal stem cells into sublethally irradiated NOD×SCID×γc$^{-/-}$ hosts[33]. Immunostaining of frozen sections of thymus grafts with anti-human CD11c revealed that human DCs were frequently associated with human CD31$^+$ blood vessels (Fig. 9a). Like their murine counterparts, some human DCs were found to breach the endothelial barrier and extend processes into the vessel lumen (Fig. 9b). Moreover, 9.4 ± 3% of human DCs in thymic grafts were stained after IV injection of anti-human CD11c-PE, and most human DCs in the spleen (56.5 ± 6.3%) and virtually all blood-borne DCs (92.9 ± 2.7%) were stained as well (Fig. 9c). Finally, upon injection of OVA-AF488, the fluorescent Ag was rapidly acquired by human DCs in the thymus (28.6 ± 3.5%) and spleen (36.2 ± 2.8%)

at a frequency that corresponded to the frequency of DCs accessible to anti-hCD11c-PE in each organ (Fig. 9d). Together, these results strongly suggest that TE-DCs occur not only in mice, but also in humans and may conceivably represent a conserved feature in thymi of all mammals.

**Discussion**

Thymic T-cell selection is critical for the development of adaptive immunity and the prevention of autoimmunity in jawed vertebrates. Positive selection, which ensures that T cells recognize self-MHC complexes, relies on Ags that are synthesized within the thymus itself[6]. In contrast, the negative selection process, which eliminates autoreactive T cells, requires a much broader spectrum of antigenic material to ensure that tolerance to innocuous Ags is reliably induced throughout the body[6]. Previous work had identified four principal pathways that ensure the presence of diverse Ags for negative selection in the thymus: (1) expression of ubiquitous self-Ags by a variety of thymus-resident cell types; (2) an ectopic expression of tissue-restricted Ags by mTECs, which depends on two transcription factors, Aire and Fezf2[12,34]; (3) transport of self (or non-self) Ags that are collected in peripheral tissues by migratory DCs, which then return to the blood via the lymphatic system and home to the thymus[16,17]; and (4) direct diffusion of small, circulating antigenic peptides across the wall of thymic microvessels[18].

Here, we identify a fifth specialized mechanism that enables efficient thymic sampling of blood-borne antigenic macromolecules that are too large to diffuse passively into the thymic parenchyma. We show that a discrete subset of DCs in murine and human thymi are uniquely positioned within the vascular wall of thymic microvessels where they project cellular processes into the bloodstream to capture circulating antigenic material. The conspicuous bi-compartmental positioning of these TE-DCs depends on interactions between a DC-expressed chemokine receptor, CX$_3$CR1, and its ligand, CX$_3$CL1, which is prominently expressed in thymic capillaries. We further show that TE-DCs capture and cross-present blood-borne proteins resulting in selective extravascular deletion of developing Ag-specific T cells.

The fact that thymic DCs play diverse roles in central tolerance induction by increasing the capacity for deletion of autoreactive T cells is well established[35–37]. Indeed, thymic DCs cross-present tissue-specific Ags generated by mTECs[38,39]. The significance of DCs in tolerance induction has been demonstrated in mutant mouse models where the absence of DCs compromised the negative selection of thymocytes[40–43]. In addition, DCs also play an important role in peripheral tolerance, i.e., the deletion or re-programming of extra-thymic, mature T cells that have escaped thymic selection[44]. Consequently, mice that lack DCs produce

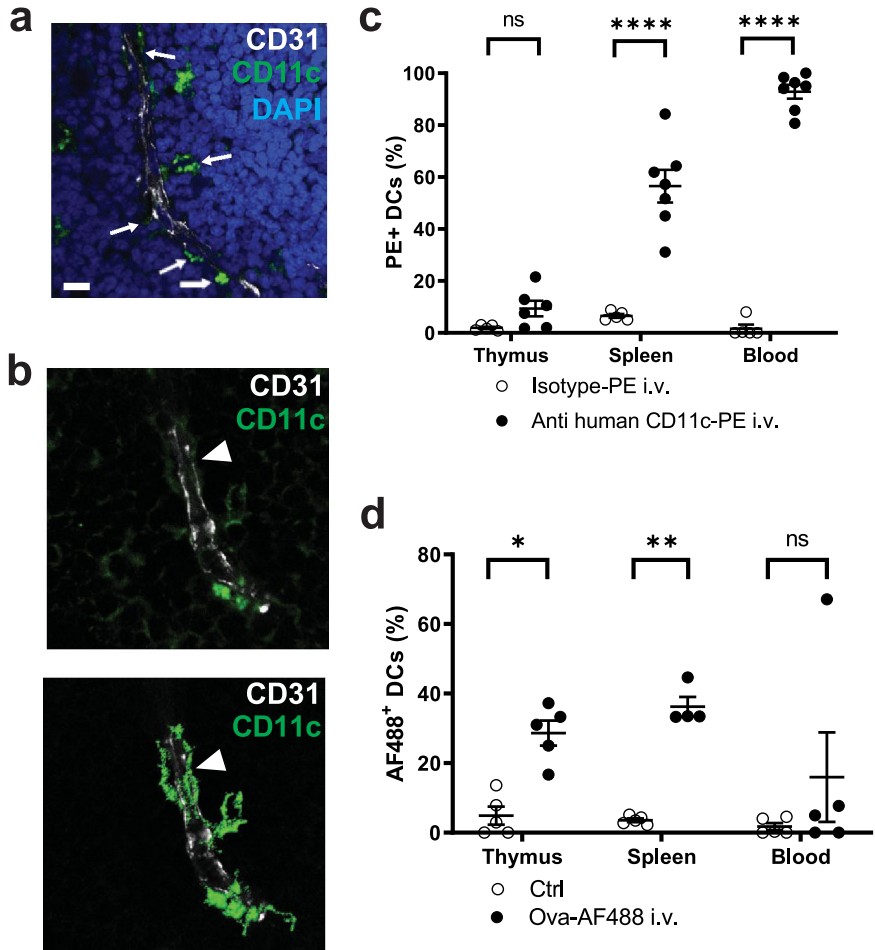

**Fig. 9 Detection of TE-DCs in human thymi. a** and **b** Representative confocal micrographs of five human thymus grafts from humanized BLT mice show the close association of human hCD11c[+] DCs (green) with hCD31[+] human, vascular endothelium (white, arrows), and nuclear DAPI staining (blue). The scale bar represents 10 µm. **b** Higher magnification images show a hCD11c[+] human DC that breaches the endothelial barrier (arrowhead). The upper panel shows hCD11c[+] DCs (green) and hCD31[+] human, vascular endothelium (white), the lower displays a surface rendered illustration of hCD11c[+] DCs. The scale bar represents 10 µm. **c** Humanized BLT mice were injected IV with anti-human hCD11c-PE or isotype-PE mAb, euthanized 2 mins later and PE[+] DCs from the animals' grafted human thymus, endogenous spleen, and blood were analyzed by flow cytometry. Each symbol represents an individual animal; open symbols indicate isotype-PE injected mice, closed symbols are anti-human CD11c-PE injected mice. $n = 5$ mice/group for isotype-PE i.v., $n = 6$ for CD11c-PE i.v. thymi and $n = 7$ for CD11c-PE i.v. spleen and blood each (two-way ANOVA). **d** Uptake of Alexa 488 labeled ovalbumin (Ova-AF488) by human DCs in BLT mice. In all, 5 min after IV injection of 100 µg Ova-AF488, single-cell suspensions of the human thymus, mouse spleen, and blood were obtained and stained for hCD11c[+] DC. $n = 5$ mice/group except for $n = 4$ for Ova-AF488 i.v. spleen (two-way ANOVA). Bars and error bars represent mean ± SEM. *$P = 0.0234$; **$P = 0.003$; ****$P < 0.0001$; n.s.: not significant.

increased numbers of autoreactive T cells resulting in symptoms of severe autoimmunity.

Thymic DCs are composed of phenotypically and functionally distinct subsets that are of either intrathymic or peripheral origin. As shown here (Fig. 1e) and reported elsewhere[27,45], the CD8α[+]CD11b[−] cDC subset comprises the majority of cDCs in the murine thymus. Most cells in this DC subset are thought to originate intrathymically from a CLP and harbor frequent D-J rearrangements, that are typically found in thymocytes[25,26,46]. Similar to CD8α[+] cDCs in the periphery, thymic CD8α[+]CD11b[−] cDCs are particularly efficient at cross-presenting exogenous Ags in MHC class I, at least when compared with other DC subsets in vitro[47,48]. In vivo, thymic cross-presentation of self-Ags by DCs contributes to the induction of CD8[+] T-cell tolerance, however, the identity of the cross-presenting DC subset(s) remain(s) to be established[49]. Of note, although the bulk of thymic TE-DCs did not express the CD8α[+]CD11b[−] phenotype, our results indicate that TE-DCs that had captured a circulating antigenic protein, OVA, were efficient at deleting OVA-specific

OT-I thymocytes both in vitro and in vivo. Since OT-I T cells are MHC class I restricted[32], it is very likely that at least some TE-DCs possess the ability to cross-present exogenous Ags in MHC class I.

Despite these functional parallels between thymic CD8α[+]CD11b[−] cDC and TE-DCs, our findings suggest that TE-DCs have an extra-thymic origin. Small numbers of cDC as well as pDCs are physiologically present in the blood and can home to the thymus[16,17]. Indeed, most intrathymic CD8α[−]CD11b[+] cDCs and pDCs are thought to originate in the BM[27,48,50,51]. These migratory DCs can acquire antigenic materials in peripheral tissues and then relocate to the thymus where they mediate deletion of autoreactive thymocytes[16,17]. The present findings suggest that circulating CD8α[−/low]CD11b[+] cDCs are also the main source of thymic TE-DCs. This conclusion is implied by our parabiosis experiments, where the level of chimerism among DCs in the blood was much higher than in thymic parenchyma, but similar to the chimerism of thymic TE-DCs (Fig. 5f–h). Moreover, the subset composition of thymic TE-DCs closely mirrored that of

splenic and blood-borne DCs and was markedly distinct from thymic Par-DCs, which are dominated by CD8a+CD11b− DCs. Thus, most, if not all, thymic TE-DCs originate in the BM and are recruited as fully differentiated DCs within thymic microvessels.

The thymic recruitment of circulating DCs as well as BM-derived CLPs is mediated by specialized microvascular ECs that line postcapillary venules and constitutively express a unique combination of adhesion molecules and chemokines to support tissue-specific multi-step adhesion cascades[16,52]. Circulating DCs initially tether to the vessel wall by engaging PSGL-1, the ligand for endothelial P-selectin (CD62P), which mediates slow rolling. Subsequently, the rolling cells arrest through binding of α4β1 integrin to endothelial VCAM-1[16]. The integrin-mediated arrest of DCs is triggered by a chemoattractant signal that depends on pertussis toxin-sensitive signaling via one or more G protein-coupled receptors. A previous study reported a role for the chemokine receptor CCR2 in the homing of pDCs and CD11b+ cDCs to the thymus[53], however, in our hands, inhibition of CCR2 did not significantly affect the short-term trafficking of adoptively transferred DCs to the thymus[16].

The present work highlights a different chemokine pathway, CX3CR1–CX3CL1, in the function of thymic CD11b+ DCs. However, we consider it unlikely that the CX3CL1-CX3CR1 pathway contributes to the multi-step adhesion cascade responsible for initial DC recruitment from the blood because in most tissues the intravascular adhesion of blood-borne leukocytes is restricted to ACKR1+ postcapillary venules[28]. Our observations in CX3CL1 reporter mice indicate that endothelial CX3CL1 expression is restricted to ACKR1− thymic capillaries (Supplementary Fig. 7), which do not support adhesion through selectins at steady state[28]. Thus, DCs more likely respond to one or more other chemoattractants, possibly including CCL2, within thymic venules. This idea is consistent with the finding that circulating CX3CR1-deficient DCs failed to establish TE-DCs, but were not compromised in their ability to home to the thymus (Fig. 6f and Supplementary Fig. 6f). The formation of cellular protrusions by extravascular CX3CR1+ myeloid leukocytes across the endothelial lining of inflamed microvessels in the brain and spinal cord of yet uncharacterized function[54] and the trans-epithelial positioning of CX3CR1+ DCs in the gut at steady state[55] likely serve as analogous immune surveillance strategies at these blood-tissue and intestinal barrier interfaces by sampling luminal antigens. Similarly, the CX3CR1+ DCs at the blood–thymus barrier sample luminal antigens and make them available to the thymic stroma for tolerance induction.

Our MP-IVM experiments indicate that many DCs that adhere within thymic venules diapedese across the vessel wall and display a high degree of motility within the extravascular parenchyma (Fig. 2c–e). It is possible that TE-DCs arise from this migratory DC population, which would require that extravascular DCs that encounter a CX3CL1+ capillary reversely diapedese across the vessel's wall and become stationary once they have partially penetrated the endothelial lining. Alternatively, some DCs that adhere in venules may also remain within the vessel lumen and crawl toward upstream microvascular segments where ECs express CX3CL1. Indeed, IVM studies in other tissues have shown that CX3CR1+ myeloid cells, particularly monocytes, actively patrol the lumen of microvessels by randomly crawling in upstream and downstream direction[56,57]. It is conceivable that newly recruited CX3CR1+ DCs engage in similar crawling activity in the thymic microcirculation to navigate out of venules and into capillaries where they may be prompted by endothelial CX3CL1 to invade the vessel wall and assume their bi-compartmental position. Using scRNA-seq on thymic CD11c+ DC subsets, Vobořil et al.[24] recently showed the expression of CX3CR1 in Xcr1–CD8α–Sirpα+ cDCs and monocyte-derived DCs. These two populations of CX3CR1+ DCs can be distinguished by their Flt3 expression, which is restricted to the cDC subset (Fig. 4h, i).

Remarkably, although chemokines are usually thought of as promoting the motility of migratory leukocytes[58], the biological effect of CX3CL1-CX3CR1 engagement in the thymus appears to cause migratory DCs to remain stationary at the blood–thymus interface for at least several days. After adoptive transfer, purified splenic DCs were found to contribute to the TE-DC pool for ~48 h (Fig. 5c, e), whereas our results in parabiotic mice combined with experiments using anti-α4 mAb to block the thymic supply of circulating DCs imply that the half-life of TE-DCs may be even as long as 1 week (Fig. 5f–j). The apparently shorter dwell time of adoptively transferred splenic DCs compared to endogenous TE-DCs may reflect differences in maturation or differentiation state, migratory properties, or survival between these DC populations. In addition, the slow decline of TE-DCs in animals treated with anti-α4 mAb could indicate that some TE-DCs may originate from progenitors within the thymus, at least in a setting where the recruitment of circulating DCs is inhibited. Regardless, our observations clearly demonstrate that TE-DCs establish a sustained presence within capillaries, whereby their intraluminal portion is continuously bathed in the blood. This long-term exposure to flowing blood presumably allows TE-DCs to sample circulating macromolecules for intracellular processing and presentation to developing thymocytes.

It is tempting to speculate that circulating Ag acquisition by TE-DCs may be particularly relevant to ensure self-tolerance towards Ags that are not encoded by the host genome. Immunological tolerance is needed for countless non-self-Ags from environmental sources, such as food and commensal bacteria. Low-level bacteremia is a common occurrence in humans, for example, as a result of chewing food or dental procedures[59]. Although there are numerous barriers that minimize microbial translocation across barrier tissues into the bloodstream[60], low-level systemic exposure to microbial products seems inevitable. Indeed, there is evidence that human tissues that are commonly presumed to be sterile, such as the spleen, contain measurable amounts of peptidoglycans from intestinal commensal flora, even in the absence of clinical infections[61]. Immunological tolerance against commensal microbiota is critical to maintaining health and immune system homeostasis[62].

One mechanism to achieve Ag-specific tolerance in the periphery is the conversion of conventional CD4 T cells into FoxP3+ regulatory T cells (Tregs)[63]. However, several lines of evidence indicate that thymus-derived natural Tregs can also recognize commensal Ags[15]. Furthermore, humans that had been vaccinated against HBV or tetanus toxoid were found to possess FoxP3+ Tregs specific for vaccine Ags. These Tregs displayed a highly demethylated region within the FOXP3 locus, indicating that they arose as natural Tregs within the thymus, presumably in response to intrathymic presentation of vaccine Ags[64]. Similarly, experiments in transgenic mice expressing membrane-bound OVA in cardiomyocytes, but not in the thymus, have documented a negative selection of OVA-specific CD4+ T cells in the animals' thymi[16,65], indicating that peripheral self-Ags can become "visible" to thymocytes even when they are not expressed within the thymus itself. The route of Ag transport in these settings has not been determined, but likely includes leakage of antigenic molecules into the bloodstream followed by capture and presentation by TE-DCs. To be sure, in some settings, blood-borne Ags can be very efficient at mediating negative thymic selection[18,66] without requiring the presence of TE-DCs. However, this activity appears to be restricted to peptides that are small enough to enter the thymus by passive diffusion (Fig. 8). The present findings identify TE-DCs as a mechanism by which antigenic macromolecules of extra-thymic origin can be made

available for negative selection and possibly also for Treg induction.

To date, TE-DCs have presumably escaped prior detection because their intimate spatial relationship with capillaries may not be apparent in histological sections or in ex vivo preparations that lack physiologic blood flow. In vitro imaging models, such as reaggregated thymic organ cultures[8] or preparations of thymic slices[7,9,10,67] have made important contributions to our understanding of thymocyte motility and interactions with Ag-presenting cells, however, these techniques require that tissues are exposed to super-physiologic ambient oxygen levels and they do not allow the study of physiologic afferent signals that may be provided by innervation or the flowing blood. On the other hand, as discussed above, intravital imaging of thoracic thymus is technically challenging due to anatomic, surgical, and optical constraints. The use of fetal thymus transplantation under the kidney capsule described here and elsewhere[68] can effectively circumvent these challenges. Consistent with prior studies, we determined that TT support normal T-cell development[69] and mirror their endogenous counterparts in terms of architecture, vascularization, and DC subset composition, localization, and homing.

A noteworthy difference between ET and TT is the fact that the medullary regions were located more superficially in the latter, presumably due to the smaller size of thymic grafts, enabling improved visualization of cellular behavior in subcortical regions. This allowed us to characterize the migratory behavior of individual homed and ET DC subsets by MP-IVM. Our observations indicate that newly homed DCs achieve velocities comparable to those of thymocytes[70] and are much more motile than thymus-resident DCs (Fig. 2c–e). Higher motility of recent DC immigrants in lymphoid tissues, as compared to tissue-resident or "old" immigrants, has also been observed in peripheral lymph nodes[71] and has been proposed as a mechanism to optimize the probability for Ag encounter by surrounding T cells.

A notable difference between DCs in the thymus vs. lymph nodes is that TE-DCs are absent from the latter despite the abundance of $CD11b^+$ DCs in lymph nodes (Fig. 4f, g). Thus, Ag sampling in lymph nodes appears to be restricted to material that enters these organs via afferent lymphatics[72]. By contrast, about half of all splenic DCs were rapidly stained by intravascular anti-CD11c-PE, consistent with the central role of the spleen in filtering the blood[73]. In addition, it should be noted that perivascular positioning has also been reported for other leukocytes, including dermal mast cells[74] and macrophages[75,76]. More broadly, transcompartmental positioning of myeloid leukocytes is thought of as a way to monitor events within specific anatomic compartments that may serve as conduits for infectious agents, such as the intestinal lumen[77–79] or the pulmonary airways[80].

In conclusion, we report the phenotypic and functional identification of a specialized DC population in murine and human thymi that form a transendothelial cellular conduit between the systemic circulation and the thymic parenchyma. This TE-DC bridge promotes the formation of central tolerance against blood-borne macromolecules that are not expressed within the thymus itself. This novel route of thymic Ag delivery is complementary, but non-redundant to at least four other established modes of thymic Ag acquisition, which together ensure the induction of central tolerance and prevention of autoimmunity.

## Methods

**Mice.** C57BL/6 mice were purchased from Charles River Laboratories (stock no: 027) and were used at 4–12 weeks of age. OT-I mice, which carry a transgenic TCR specific for ovalbumin amino acids 257–264 (SIINFEKL) in H-2K$^b$, were from Taconic Farms (stock no: 2334)[81]. Cx3cr1$^{gfp/gfp}$ × CD45.1 (stock no: 008451), Cx3cr1$^{gfp/gfp}$ × CD45.2 (stock no: 005582) and CD11c-YFP mice (stock no:

008830) were purchased from the Jackson Laboratory and are described elsewhere[20,82]. Cx3cr1$^{gfp/gfp}$ × OT-I × CD45.2 mice were generated in the lab by crossing Cx3cr1$^{gfp/gfp}$ × CD45.2 (stock no: 005582) to OT-I mice (stock no: 2334)[81,82]. Cx3cl1$^{-/-}$ and Cx3cl1-mCherry mice were a gift from A. Lira and S. Jung, respectively[83,84]. Mice were housed in a specific pathogen-free and viral antibody-free animal facility. Control and experimental mice were co-housed. All mice were female and 4–12 weeks of age unless otherwise stated. Mice were housed in 12/12 hrs light/dark cycles (6 am to 6 pm light) at a 70~72 F degree temperature range, and relative humidity within 40~50%.

**Reagents.** Commercially available unconjugated and fluorochrome-conjugated Abs are listed in Supplementary Table 1. Chimeric, non-depleting, anti-α$_4$ integrin mAb (CRL19.11) was provided by R. Palframan. Anti-cytokeratin 8 mAb secreting hybridoma (TROMA-1) was obtained from the Developmental Studies Hybridoma Bank at the University of Iowa and PTX was from Calbiochem. Alexa Fluor® 488, Alexa Fluor® 647-labeled ovalbumin was purchased from Invitrogen and unlabeled EndoGrade ovalbumin was from Profos. High molecular weight fluorescein iso-thiocyanate and TRITC-dextran (MW = $2 \times 10^6$ Da) were from Invitrogen. Fluorescently labeled Ulex Europaeus Agglutinin I was purchased from Vector Laboratories.

**Cell preparation.** To analyze DCs in thymi, LNs, and spleens, the organs were harvested and digested in 50 µg/ml (for thymi and LNs) or 100 µg/ml Liberase TM (for spleens; Roche) for 20 min at 37 °C. Single-cell suspensions for analysis of lymphocytes in thymi, LNs, and spleens were generated by mechanical organ dissociation. Blood samples were obtained by retroobital bleeding followed by red blood cell lysis in ACK buffer for 60 s. Cells were stained for surface markers and analyzed on a FACS Canto (BD Biosciences). For cell sorting, thymi were digested in 50 µg/ml Liberase TM (Roche) for 20 min at 37 °C followed by density-gradient centrifugation. Thymic stromal cells preparation was performed by washing and resuspending cells in Percoll (ρ, 1.115; GE Healthcare). A discontinuous gradient was then generated by the addition of a layer of Percoll (ρ, 1.050) followed by a layer of PBS on top of this cell suspension. Gradients were spun for 30 min at $1350 \times g$ at 4 °C, and low-density cells were collected from the upper interface, washed, and stained for sorting by flow cytometry. DCs were prepared by centrifugation over NycoPrep (Axis-Shield) according to the vendor's manual. The FACS gating strategies are depicted in Supplementary Fig. 8.

**Homing assays.** Donor DCs were isolated from C57BL/6 mice that had been injected subcutaneously with $2 \times 10^6$ to $5 \times 10^6$ B16 melanoma cells secreting Flt3 ligand as described[85]. After 10–14 d, mice were killed and splenic DCs were purified by digestion in 100 µg/ml Liberase TM (Roche) for 20 min at 37 °C followed by density-gradient centrifugation over NycoPrep (Axis-Shield) according to the vendor's manual. These preparations routinely contained 75–85% $CD11c^+$ DCs. When indicated, DC maturation was induced by culture for 24 h in complete medium in the presence of 1 µg/ml of lipopolysaccharide (E. coli 0.26:B6; Sigma-Aldrich). Mature DC cultures typically resulted in enrichment in $CD11c^+$ cells (90–95%) and in the upregulation of classical maturation markers (CD86 and MHC class II) for all $CD11c^+$ cells. Immature or mature DCs were labeled for 15 min at 37 °C with 30 µM CFSE (carboxyfluorescein succinimidyl ester) or 3 µM DDAO-SE (7-hydroxy-9H-(1,3-dichloro-9,9-dimethylacridin-2-one) succinimidyl ester (all from Molecular Probes) at a cell concentration of $10^7$ cells/ml. Dead cells and excess labels were removed by centrifugation, and $1 \times 10^7$ labeled DCs were then injected into the tail veins of recipient mice. In some experiments, DCs from Cx3cr1$^{gfp/gfp}$ mice were isolated or WT DCs were pretreated with 50 µg/ml of blocking mAb or PTX (Calbiochem) for 1 hour at 37 °C and washed before simultaneous injection with control populations; for inhibition of endothelial adhesion molecules, 100 µg mAb was injected along with the labeled DCs. Mice were killed at indicated time points and single-cell suspensions were generated from spleens and thymi and cell samples were analyzed by flow cytometry. The total number of homed DCs was calculated by multiplication of the fraction of $CFSE^+$ (or DDAO-SE$^+$) $CD11c^+$ events by the total cellularity of the target organ.

Homing of DC subsets and stability of DC subsets were addressed by transfer of DC subsets that had been sorted on a FACS Aria II (BD Bioscience) followed by fluorescent labeling with CFSE, DDAO-SE, and CMTMR, respectively (all from Molecular Probes) and intravenous injection into recipient mice. Mice were killed by $CO_2$ asphyxiation followed by cervical dislocation and homed DCs were analyzed for their respective subset 18 h after transfer on an LSR II (BD Biosciences). The FACs gating strategies are depicted in Supplementary Fig. 8.

**Bone marrow transplantation.** In all, $1–5 \times 10^7$ CD4 and CD8 depleted BM cells from the tibia and femur from donor animals were injected into the lateral tail vein of lethally irradiated C57BL/6 mice (myeloablative, $2 \times 650$ rad, Cs source irradiation).

Thymus transplantation and MP-IVM. Individual thymus lobes from E14.5–18.5 embryos were transplanted under the kidney capsule of anesthetized, male recipient mice. On the day of imaging, TT was gently exposed and immobilized on a Styrofoam platform with two 27G1/2 needles. The exposed thymus was submerged in sterile saline and covered with a glass coverslip. A

thermocouple was placed next to the thymus to monitor local temperature, which was maintained at 37–39 °C. MPM was performed on a Prairie system at an excitation wavelength of 850 nm, from a tunable MaiTai Ti:sapphire laser (Spectra-Physics). For three and four-dimensional offline analysis of thymic architecture and cell migration, stacks of 9-11 optical x–y sections with 3 μm z spacing were acquired every 15 s or 20 s, respectively, with physical zooming to ×1–2 through a ×20/0.95 numerical aperture water-immersion objective lens (Olympus). Emitted fluorescence and second-harmonic signals were detected through 455/70 nm, 450/80 nm, 590/50 nm, and 665/65 nm bandpass filters with non-descanned detectors to generate four-color images. Sequences of image stacks were transformed into volume-rendered three-dimensional volumes.

**Image analysis**. T series were converted into movies using Volocity (Perkin Elmer) and Imaris (Bitplane) software. Motility parameters such as three-dimensional instantaneous velocities, mean velocities per track, mean displacement plots, meandering indices, and motility coefficients were determined by semi-automated cell tracking with Imaris and Volocity Software and computational analysis by MatLab (Mathworks), Excel (Microsoft Office), or Prism (GraphPad). The meandering index is a measure for the directionality of cell migration; it represents the ratio of the displacement over the total path length of the track. The motility coefficient measures a cell's propensity to move away from its point of origin, analogous to the diffusion coefficient of Brownian motion.

**Immunofluorescence**. For cryosections, thymi were harvested, fixed in phosphate-buffered L-lysine with 1% paraformaldehyde/periodate, dehydrated in 30% sucrose in PBS, snap-frozen in TBS tissue-freezing liquid (Triangle Biomedical Sciences), and stored at −80 °C. Sections of 30 μm thickness were mounted on Superfrost Plus slides (Fisherbrand) and stained with fluorescent antibodies in a humidified chamber after Fc-receptor blockade with 1 μg/ml antibody 2.4G2 (Bio X cell). Samples were mounted in FluorSave reagent solution (EMD-Calbiochem) and stored at 4 °C until analysis. Images were collected with a Bio-Rad confocal microscopy system using an Olympus BX50WI microscope and ×10/0.4 numerical aperture or ×60/1.2 numerical aperture water-immersion objective lenses. Images were analyzed with LaserSharp2000 software (Bio-Rad Cell Science), Volocity (Perkin Elmer), and Imaris (Bitplane).

Intravascular labeling, calculation of staining indexes, determination of Lu, Par, and TE localization. Leukocytes were labeled intravascularly by IV injection of 1 μg PE-conjugated mAb to CD45 or CD11c, or injection of fluorescently labeled ovalbumin. Animals were euthanized by CO₂ asphyxiation followed by cervical dislocation 2 minutes post injection of mAb or 5 minutes post injection of ovalbumin by CO₂ asphyxiation. Staining indexes were generated by competitive in vivo and ex vivo staining with anti-CD11c-PE (for in vivo, clone N418) and anti-CD11c-PE-Cy7 (for ex vivo staining, clone N418) and ex vivo staining with anti-hamster-AF647 mAb to label all CD11c⁺ cells. Each cell's FI for the in vivo and the ex vivo label, respectively, was divided by the FI for anti-hamster-AF647 using FCS Express (De Novo Software). The resulting values for in vivo and ex vivo staining indexes were graphed and analyzed using Prism (GraphPad) and used to evaluate the cells' specific localization. Blood-borne DCs were used to determine luminal localization and thymic DCs from isotype-PE injected animals for parenchymal localization. TE cells were defined as cells with an intermediate phenotype compared with Lu and Par-DCs.

**Chemotaxis assay**. Chemotaxis assays were carried out in 24-well transwell plates with 5 μm pore size (Costar, Cambridge, MA). CD11c⁺ thymic DCs were isolated by positive selection using MACS sorting (Miltenyi) according to the manu-facturer's manual. In all, 10⁷ thymic DCs were loaded into the upper chamber of the transwell plate. The lower chamber contained media with or without mouse recombinant CX₃CL1 (R&D). Transwells were incubated for 3 h at 37 °C; input cells and cells in the lower chamber were counted and analyzed by FACS. Data were presented as chemotactic index and percent of migrated cells of input cells. The chemotactic index was calculated as the ratio of migrated cells versus back-ground migration in the absence of a chemokine.

**In vitro deletion assay**. Thymic DCs were isolated from untreated or ovalbumin injected animals. Bulk CD11c⁺ cells were isolated by positive selection using MACS sorting (Miltenyi) according to the manufacturer's manual. Responder cells were isolated from thymi from OT-I TCR tg animals by positive selection for CD4⁺ cells using MACS sorting according to the vendor's manual. DCs and thymocytes were co-cultured for 3 days at a DCs:T cells ratio of 10:1. In vitro deletion was assessed by flow cytometry.

**In vivo deletion assay**. Lethally irradiated (myeloablative, 2 × 650 rad, Cs source irradiation) CD45.1 or CD45.1 × Cx3xr1ᵍᶠᵖ/ᵍᶠᵖ recipient mice were reconstituted with 1:1 mixture of CD45.2 × OT-I and CD45.2 or CD45.1 × Cx3xr1ᵍᶠᵖ/ᵍᶠᵖ and CD45.1 × Cx3xr1ᵍᶠᵖ/ᵍᶠᵖ × OT-I BM. Donor BM was isolated from the tibia and femur and CD4⁺ and CD8⁺ cells were depleted using MACS sorting. Lethally irradiated recipient mice were injected with 1–5 × 10⁷ BM cells into the lateral tail vein. In all, 18 days after BM transfer, BM chimeric animals were injected IV with

anti-α₄ mAb in combination with 100 μg ovalbumin or 10 μg SIINFEKL peptide, or left untreated. The deletion was assessed by FACS analysis 24 h later.

**Humanized mice**. NOD/SCID/IL2Rγ⁻/⁻ (NODSCIDGAMMA–NSG) mice (The Jackson Laboratory) were housed in a specific pathogen-free facility at Massachusetts General Hospital.

BLT mice generation was performed as previously described[33]. In brief, 6–8-week-old NSG mice were whole-body irradiated by a sub-lethal dose (2 Gy), anesthetized, and implanted with 1-mm³ fragments of human fetal thymus and liver tissue in a pocket created under the murine kidney capsule. Human fetal tissues (17–19 weeks of gestational age) were made available through Advanced Bioscience Resources (ABR; Alameda, CA). A total of 1 × 10⁵ autologous fetal liver tissue-derived human CD34+ hematopoietic stem cells were then injected intravenously in a tail vein within 6 h of tissue transplantation. Mice were maintained in micro isolator cages and fed autoclaved food and acidified water. Human immune reconstitution was then monitored over 12–14 weeks. Mice were generally considered reconstituted if >50% of cells in the lymphocyte gate were human CD45⁺ and >40% of these human CD45⁺ lymphocytes were human CD3⁺.

**Statistical analysis**. Normally distributed data sets are presented as mean ± SEM. Statistical significance was assessed by a two-tailed Student's t test for comparison of two groups or by one-way or two-way analysis of variance followed by the Student Newman Keuls post-test for more than two groups. Non-normally dis-tributed data were presented as median and compared with Mann–Whitney U test (two groups) or the Kruskal–Wallis test followed by Dunn's test to compare multiple samples. Differences with P values of <0.05 were considered statistically significant.

**Ethics statement**. Mice were housed in a specific pathogen-free and viral antibody-free animal facility. All experiments were approved by the Institutional Animal Care and Use Committee (IACUC) at Harvard Medical School (Harvard Medical Area (HMA) Standing Committee on Animals) and performed under the approved protocols IS00000725-3 and IS00000159-3. Anonymized human fetal tissues were acquired through ABR (Alameda, CA) and utilized under a Massa-chusetts General Hospital Institutional Review Board-approved protocol. All humanized mouse experiments were approved by the Massachusetts General Hospital Institutional Animal Care and Use Committee (IACUC) and performed under the approved protocol 2009N000136. All the human tissue used in the experiment for the implantation in the BLT mice was obtained from consenting donors. Consent forms are obtained by the clinics performing the procedure. MGH IRB Committee approved the use of human tissue.

**Software and code**. No commercial, open-source, or custom code was used to collect the data in this study. For data analysis, Flowjo version 10.6.1 and Imaris version 9.5.0 were used.

**Biological material**. The anti-Darc antibody generated by the lab is readily available from the authors to researchers upon request.

**Reporting summary**. Further information on research design is available in the Nature Research Reporting Summary linked to this article.

## Data availability

The raw and processed data that support the findings of this study are available from the corresponding author upon reasonable request. Source data are provided with this paper.

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

## Acknowledgements

We thank A. Lira for kindly providing Cx3cl1$^{-/-}$ mice. This work was supported by NIH grants PO1 AI112521 and R01 AI155865 (to Ulrich H. von Andrian), R01DK126753 (to Ki-wook Kim), DFG (Deutsche Forschungsgemeinschaft) Research Fellowship (to Kristin Rattay, GZ: RA 2984/1-1) and the HMS Center for Immune Imaging.

## Author contributions

E.H.V. and U.H.v.A. conceived the study. E.H.V. designed and performed experiments and analyzed the data. A.T. performed in vivo experiments, R.A.F. conducted and analyzed in vivo CD11c-PE-labeling studies and V.V. performed and analyzed the humanized mouse experiments. O.B. analyzed and interpreted the 3D rendering of the TE-DC protrusion microscopy data and performed FACS analysis for Fig. 7. K.K. and S.J. provided the CX3CL1-mCherry mice. A.M.T. provided humanized mice. E.H.V., K.R., O.B., and U.H.v.A. wrote the manuscript with input from A.T.

## Competing interests

The authors declare no competing financial or non-financial interests.
