## [Peer Review File · Nature Communications]

REVIEWER COMMENTS

Reviewer #1 (DC-T interaction, T cell biology) (Remarks to the Author):

Summary

This is a high quality manuscript, containing many carefully-analysed and elegant experiments. The authors build on their previous work with showing that dendritic cells from blood migrate to the thymus. Here they argue that these dendritic cells reside in the interface between the vessel wall and the thymus parenchyma, and extend transepithelial dendrites. These dendrites enable capture of proteins, which can then be cross-presented for deletion of Ag-specific thymocytes. This behaviour depends on the expression of CX3CR1 by the largely CD11c⁺ CD8⁻ CD11b⁺ dendritic cell population.

Major Comments

While the data are well-described and presented, I am not yet persuaded by all aspects of the work. Specifically:

The authors go to some length to persuade the reader that the TE-DCs are bona-fide dendritic cells, rather than a circulating monocyte-derived population. Their arguments here are not fully convincing. The key data (Fig 2K,I, and associated supplementaries) lack positive controls to show the expression of these markers in the relatively numerous cDC population that should have been accessible to the authors. They also seem to contradict recently-published ssSEQ data indicating that the here-ignored monocyte-derived DCs in the thymus express the highest levels of CX3CR1 (<https://www.nature.com/articles/s41467-020-16081-3/figures/5>)

Very few of the provided figures provide data about the absolute numbers of cells described, being almost exclusively presented as proportions of transferred cells. For instance, in Figure 1 "Cumulative frequency" is as % of max. So obscures relative numbers of populations. Providing absolute numbers would be very helpful, or at least proportions relative to a reference population (e.g. CD45⁻ cells), or even to an area/volume for microscopy. Similarly, the data in Figure 3 could be much better interpreted if the reader were told the absolute numbers of thymus "DCs" under consideration at each of the timepoints. These data may not change the authors' conclusions, and could be provided in supplementary information, but would reveal, for instance, whether the thymus populations of interest were proliferating or reducing in numbers during the course of the experiments. Similarly, it would be important to know the numbers of cells populating the thymus after transfer of 10^7 cells. As it stands the efficiency of this repopulation is unclear - and this is important if we are to consider whether the observed phenomena are physiologically-important. In light of the above, to support the authors' conclusions, the absolute numbers of cells described in their key experiments should be provided.

Minor Comments

If the authors wish to claim that the TE-DCs are Flt3 dependent, they could consider testing this directly.

From the data shown in Fig 5b it is difficult to understand how the percentages in 5C are generated. Representative plots for the other conditions should be shown.

Comments provided by Simon Milling.

Reviewer #2 (T cell activation/signaling, immune imaging) (Remarks to the Author):

In "Specialized transendothelial dendritic cells mediate thymic T cell selection against blood-borne

macromolecules", Vollman et al. characterize a population of CD11b+CD8- thymic dendritic cells (DCs) that exhibit sustained, partial exposure to vascular contents in the thymus. Using gene knockouts, intravital imaging, and functional tracking of protein antigen, they provide fairly compelling evidence that this population of transendothelial (TE) DCs captures protein antigen from the blood for presentation to thymocytes. They do not show whether this new pathway of thymic antigen presentation plays a physiologically important role in shaping the T cell repertoire. Taking this next step, however, is arguably beyond the scope of the present manuscript, which is quite interesting already. My concerns as a reviewer are largely technical, and I imagine the authors will be able to address them without too much trouble.

1) My strongest reservations have to do with the presentation and analysis of imaging data.

A) More must be done to define precisely how the TE-DCs sample the vascular lumen. Do they form dedicated sampling structures or simply extravasate and intravasate repeatedly? What do the transendothelial structures look like (higher magnification/resolution imaging would be helpful here)? It would also be good to know what fractions of thymic DCs display transendothelial projections, the average size of the projections, and their lifetimes.

B) The authors should confirm histologically that the CD11b+CD8- TE-DC population and the DCs that cling to blood vessels and presumably sample vascular contents are one and the same. This would enable them to map their flow cytometric findings back onto the tissue.

C) The criticisms described in (A) and (B) also apply to the imaging studies of human tissue shown in Figure 7.

D) Finally, the staining image shown in Figure S7 is not compelling on its own. The results must be quantitated.

2) I am perplexed by the bottom two bars in Fig. 4F, which appear to indicate that Cx3cr1 KO DCs occupy the PE+ thymic DC niche to the same extent as WT DCs. Do PE+ DCs represent TE-DCs in this experiment, and if so, why do the KO DCs and WT DCs behave similarly? If not, could the TE DCs be included in this graph?

3) The flow cytometric analysis in Fig 5B-C is not compelling. The authors should use Figure 5B (with multiple plots, if necessary) to highlight the populations they quantify in 5C. The populations indicated in the current 5B plot appear to be noise.

Related to this experiment, perhaps there would be better overlap between the OVA+ and PE+ TE-DC populations if the authors injected the OVA protein a few hours earlier, and then followed with the CD11c-PE? This would give the TE-DCs time to engulf the protein antigen.

4) On p15, paragraph 2, the authors should emphasize that the OT1 donors used in Figure 6C and 6D were also Cx3cr1GFP/GFP (Am I correct?). This is not clear from the text (or the figure), and unless I am mistaken, it makes a big difference.

Point-by-point reply to the reviewers' comments

We thank the reviewers for their constructive comments and suggestions on our manuscript. We changed the text and figures accordingly where applicable and refer to those changes in our point-by-point reply below. The reviewer comments are highlighted in boxes, our reply is given below the respective box. Furthermore, to facilitate the re-review of our manuscript, we have **highlighted** all relevant changes in the text.

REVIEWER COMMENTS

Reviewer #1:

This is a high quality manuscript, containing many carefully-analysed and elegant experiments. The authors build on their previous work with showing that dendritic cells from blood migrate to the thymus. Here they argue that these dendritic cells reside in the interface between the vessel wall and the thymus parenchyma, and extend transepithelial dendrites. These dendrites enable capture of proteins, which can then be cross-presented for deletion of Ag-specific thymocytes. This behaviour depends on the expression of CX3CR1 by the largely CD11c+ CD8- CD11b+ dendritic cell population.

We thank the reviewer for this overall positive assessment of our work.

1) Major Comments

While the data are well-described and presented, I am not yet persuaded by all aspects of the work. Specifically:

The authors go to some length to persuade the reader that the TE-DCs are bona-fide dendritic cells, rather than a circulating monocyte-derived population. Their arguments here are not fully convincing. The key data (Fig 2K,I, and associated supplementaries) lack positive controls to show the expression of these markers in the relatively numerous cDC population that should have been accessible to the authors. They also seem to contradict recently-published ssSEQ data indicating that the here-ignored monocyte-derived DCs in the thymus express the highest levels of CX3CR1 (<https://www.nature.com/articles/s41467-020-16081-3/figures/5>)

We thank the reviewer for pointing out the recent publication by Vobořil et al. on TLR signaling in thymic epithelium and its effect on moDC recruitment (*Nature Communications*, 2020). We agree that the referenced figure 5 is of particular importance in the context of our study. We now cite this reference in the Results (page 11, line 307-310) and Discussion (page 20, line 584-588) sections of our revised manuscript.

Vobořil et al. performed single cell RNA-seq on FACS sorted Gr-1-CD11c+TdTOM+ DCs from the thymus of Foxn1CreROSA26TdTOMATO mice. Figure 5b of that paper shows a gene signature analysis for thymic DC, cDC1, cDC2, pDC, monocyte/macrophage and monocyte populations, which indicates that *Cx3cr1* mRNA expression levels are highest in the moDC population, followed by cDC2s. Since our results indicate that CX3CR1 is critical for thymic TE-

DC localization, it is important to determine whether TE-DCs are composed of the cDC2 subset (as we had originally proposed) or moDC or both.

According to the scRNA-seq data in Fig. 5b of the study by Vobořil et al., only thymic cDCs, but not moDCs or pDCs express Flt3. Our analysis in Figure 2k and l of our manuscript shows that Flt3 is uniformly expressed on thymic TE-DCs. Thus, even though moDCs express *Cx3cr1* (at least at the mRNA level), the absence of Flt3 on this DC population argues against a substantial contribution by moDCs to the TE-DC subset. This point is now discussed on pp. 11 and 20 of our paper.

2) Very few of the provided figures provide data about the absolute numbers of cells described, being almost exclusively presented as proportions of transferred cells.

For instance, in Figure 1 “Cumulative frequency” is as % of max. So obscures relative numbers of populations. Providing absolute numbers would be very helpful, or at least proportions relative to a reference population (e.g. CD45⁻ cells), or even to an area/volume for microscopy.

Similarly, the data in Figure 3 could be much better interpreted if the reader were told the absolute numbers of thymus “DCs” under consideration at each of the timepoints. These data may not change the authors conclusions, and could be provided in supplementary information, but would reveal, for instance, whether the thymus populations of interest were proliferating or reducing in numbers during the course of the experiments.

Similarly, it would be important to know the numbers of cells populating the thymus after transfer of 10⁷ cells. As it stands the efficiency of this repopulation is unclear - and this is important if we are to consider whether the observed phenomena are physiologically-important.

In light of the above, to support the authors’s conclusions, the absolute numbers of cells described in their key experiments should be provided.

We agree that quantitative information on absolute numbers of thymic DC subsets would be desirable. Nonetheless, we chose to display data obtained from FACS experiments of thymic single-cell suspensions as percentages because of the effect that the isolation procedure has on cell recovery. The need for enzymatic and mechanic tissue dissociation can substantially affect the yield of viable cell populations, and this confounding effect varies between different tissues and leukocyte populations. This unavoidable artifact, which can render absolute cell numbers highly unreliable, but has only a minimal impact on relative cell frequencies, was discussed in detail by Steinert, E. M., et al. ("Quantifying Memory CD8 T Cells Reveals Regionalization of Immunosurveillance." *Cell* 161(4): 737-749 (2015)).

Additionally, because the vast majority of cells in unfractionated thymus are thymocytes and T-cells, we performed density gradient purification of thymic stromal cells and DCs in order to enrich for these relatively rare cell populations (as described in Materials & Methods).

It is certainly feasible to provide information on the absolute yield of DCs in individual experiments, but the data would still have to be normalized to the recovered total cell count and not display 'true' numbers. Therefore, we prefer to display percentages as arguably the most interpretable way to report our data. Nevertheless, for the reviewer's information, we provide below an example of the absolute cell numbers for experiments in Figure 1 F-J, which illustrate the degree of repopulation of transplanted vs. endogenous thymi:

	Total cellularity \pm STDEV	Total DC counts \pm STDEV
Endogenous thymus	1.50E+08 \pm 2.37E+07	2.95E+06 \pm 8.57E+05
Spleen	1.67E+08 \pm 6.98E+06	6.76E+06 \pm 1.84E+06
Transplanted Thymus	9.12E+07 \pm 1.22E+07	1.27E+06 \pm 4.35E+05

3) Minor Comments

If the authors wish to claim that the TE-DCs are Flt3 dependent, they could consider testing this directly.

We used Flt3 in combination with c-Kit and CD26 as markers to distinguish thymic DCs from macrophages, but we did not specifically test the role of Flt3 on TE-DC localization or function nor do we claim that TE-DCs are Flt3 dependent. We agree that this might be an interesting future direction to pursue, but this question is beyond the scope of the current manuscript.

4) From the data shown in Fig 5b it is difficult to understand how the percentages in 5C are generated. Representative pots for the other conditions should be shown.

We agree that our approach in the original Fig. 5b to present results as a contour plot was suboptimal and we thank the reviewer for pointing this out to us. We have revised that figure and now depict the data by showing a contour plot combined with single dots representing individual events. Additionally, we now identify population names in each quadrant and also include subset frequencies (%) in the figure. Furthermore, we have reanalyzed our results in Fig.5c, which summarizes data from 5 mice/group that were carefully re-analyzed as shown in Fig. 5B and compared using a more appropriate statistical test. Further details are provided in the revised legend for Figure 5 and on page 14, lines 408-411.

Reviewer #2:

In "Specialized transendothelial dendritic cells mediate thymic T cell selection against blood-borne macromolecules", Vollmann et al. characterize a population of CD11b+CD8- thymic dendritic cells (DCs) that exhibit sustained, partial exposure to vascular contents in the thymus. Using gene knockouts, intravital imaging, and functional tracking of protein antigen, they provide fairly compelling evidence that this population of transendothelial (TE) DCs captures protein antigen from the blood for presentation to thymocytes. They do not show whether this new pathway of thymic antigen presentation plays a physiologically important role in shaping the T cell repertoire. Taking this next step, however, is arguably beyond the scope of the present manuscript, which is quite interesting already. My concerns as a reviewer are largely technical, and I imagine the authors will be able to address them without too much trouble.

1) My strongest reservations have to do with the presentation and analysis of imaging data. A) More must be done to define precisely how the TE-DCs sample the vascular lumen. Do they form dedicated sampling structures or simply extravasate and intravasate repeatedly? What do the transendothelial structures look like (higher magnification/resolution imaging would be helpful here)?

We agree with the reviewer that a further characterization of the transendothelial protrusions formed by TE-DCs could help to clarify the mechanism and kinetics of protrusion formation and antigen sampling. In theory, this analysis should be possible by using high resolution intravital microscopy in transplanted thymi. Unfortunately, although our MP-IVM setup allows us to detect individual thymic DCs that appear to be TE-DCs (as exemplified in **Suppl. Movie 3**), the resolution of these *in vivo* recordings is not sufficient for fine mapping of the blood exposed vs. extravascular aspects of individual cells.

However, we have performed a new high-resolution 3D analysis of serial confocal micrographs of frozen thymic sections in order to measure the average size of thymic TE-DC protrusions. This improved imaging strategy allows realistic digital surface rendering of thymic DCs and anti-CD31 labeled microvessels (new **Fig. 2d,e** and **Suppl. Movies 4 and 5**) as well as measurements of intra- and extravascular aspects of TE-DCs. However, the precise structural dimensions that can be derived from this image processing approach depend, at least in part, on the chosen image rendering parameters, i.e. the apparent size of both the vessel lumen and DC boundaries may be somewhat larger or smaller depending on the stringency of applied rendering rules. Thus, we have measured TE-DC protrusions with both low and high stringency rendering settings. The range of measured dimensions (expressed as % surface area per DC) provides an upper and lower boundary of the true size of TE-DC protrusions, which is now described on p. 8, lines 205-219 of the revised manuscript.

It would also be good to know what fractions of thymic DCs display transendothelial projections, the average size of the projections, and their lifetimes.

The DC subset composition and frequency of thymic Par-DC and TE-DCs as well as blood-borne, splenic and LN DCs was provided in **Figure 2j**.

B) The authors should confirm histologically that the CD11b+CD8- TE-DC population and the DCs that cling to blood vessels and presumably sample vascular contents are one and the same. This would enable them to map their flow cytometric findings back onto the tissue.

Histological staining of thymic sections for CD8 is difficult to interpret because this marker is widely expressed on CD8+ SP and DP thymocytes. The abundance of these T cell populations makes it extremely difficult to discern between CD8+ and CD8-neg DCs. These limitations of histological staining strategies prompted us to resort to FACS analysis, as this method allows us to identify and analyze cell populations of interest much more accurately and reproducibly.

C) The criticisms described in (A) and (B) also apply to the imaging studies of human tissue shown in Figure 7.

Please see our reply to comments A-B above.

D) Finally, the staining image shown in Figure S7 is not compelling on its own. The results must be quantitated.

We apologize that the original version of this supplementary figure (which makes a relatively minor point) was not sufficiently compelling. We realize that the micrographs in the original figure were too small and insufficiently labeled to optimally confer the information as intended. We have reformatted

this figure to more clearly illustrate that ACKR1 and CX3CL1 are expressed in distinct segments of the thymic microvascular network.

2) I am perplexed by the bottom two bars in Fig. 4F, which appear to indicate that Cx3cr1 KO DCs occupy the PE+ thymic DC niche to the same extent as WT DCs. Do PE+ DCs represent TE-DCs in this experiment, and if so, why do the KO DCs and WT DCs behave similarly? If not, could the TE DCs be included in this graph?

We apologize for this error. In the course of the figure editing a mix-up of the Y-axis labelling occurred. We would like to thank the reviewer for catching this mistake. The erroneous label in the figure panel has been corrected.

3) The flow cytometric analysis in Fig 5B-C is not compelling. The authors should use Figure 5B (with multiple plots, if necessary) to highlight the populations they quantify in 5C. The populations indicated in the current 5B plot appear to be noise.

Related to this experiment, perhaps there would be better overlap between the OVA+ and PE+ TE-DC populations if the authors injected the OVA protein a few hours earlier, and then followed with the CD11c-PE? This would give the TE-DCs time to engulf the protein antigen.

We would like to thank the reviewer for the suggestion of an alternative way of plotting the data in Fig 5B. A similar concern was also expressed by Reviewer #1. As explained above, we explored different ways to depict the data and plotted the events in the gates as individual dots (contour plot plus individual events), instead of plotting the events as contour plot only. Additionally, we now show the population frequencies for each gate in Fig.5b in each quadrant. Further, we reanalyzed Fig.5c based on one of the experiments (n=5) that was re-gated and reanalyzed using a more appropriate statistical test. We also changed the Figure legend accordingly and have revised the corresponding passage in Results (p. 14, lines 408-411).

4) On p15, paragraph 2, the authors should emphasize that the OT1 donors used in Figure 6C and 6D were also Cx3cr1GFP/GFP (Am I correct?). This is not clear from the text (or the figure), and unless I am mistaken, it makes a big difference.

We thank the reviewer for pointing out the Figure 6b and c labeling. We changed the Figure legend. The new description distinguished the two experimental setups more clearly (p. 36, lines 1043-1045):

(b) Lethally irradiated CD45.1 animals were reconstituted with mixed BM at a 1:1 ratio of CD45.2 and OT-I × CD45.1/2

and

(c) Lethally irradiated CX₃CR1^{-/-} × CD45.1 animals were reconstituted with mixed BM at a 1:1 ratio of CD45.1 × CX₃CR1^{-/-} and OT-I × CD45.1/2 × CX₃CR1^{-/-} BM

REVIEWERS' COMMENTS

Reviewer #1 (Remarks to the Author):

I thank the authors for their responses to my comments, and have no additional questions.

Signed; Simon Milling

Reviewer #2 (Remarks to the Author):

The authors have done a very good job at addressing my concerns.

I do not mean to belabor this point, but I really do think that the paper would be more effective if it were explicitly stated in both the text and the legends that OT1 CX3CR1^{-/-} bone marrow was used for the experiment in 6C.

point-by-point response to the reviewers' comments

REVIEWERS' COMMENTS

Reviewer #1 (Remarks to the Author):

I thank the authors for their responses to my comments, and have no additional questions.

Signed; Simon Milling

Reviewer #2 (Remarks to the Author):

The authors have done a very good job at addressing my concerns.

I do not mean to belabor this point, but I really do think that the paper would be more effective if it were explicitly stated in both the text and the legends that OT1 CX3CR1^{-/-} bone marrow was used for the experiment in 6C.

Author's response: We revised the manuscript text and now provide this information in the Figure legend and manuscript text.